# CREDAL WRAPPER OF MODEL AVERAGING FOR UNCERTAINTY ESTIMATION IN CLASSIFICATION

**Kaizheng Wang**[1,4,5,*] **Fabio Cuzzolin**[3] **Keivan Shariatmadar**[2,4,5] **David Moens**[2,5] **Hans Hallez**[1,4]

[1]DistriNet, Department of Computer Science, KU Leuven, Belgium
[2]LMSD, Department of Mechanical Engineering, KU Leuven, Belgium
[3]School of Engineering, Computing and Mathematics, Oxford Brookes University, UK
[4]M-Group, KU Leuven, Belgium     [5]Flanders Make@KU Leuven, Belgium
`kaizheng.wang@kuleuven.be`  `fabio.cuzzolin@brookes.ac.uk`
`{keivan.shariatmadar, david.moens, hans.hallez}@kuleuven.be`

## ABSTRACT

This paper presents an innovative approach, called *credal wrapper*, to formulating a credal set representation of model averaging for Bayesian neural networks (BNNs) and deep ensembles (DEs), capable of improving uncertainty estimation in classification tasks. Given a finite collection of single predictive distributions derived from BNNs or DEs, the proposed credal wrapper approach extracts an upper and a lower probability bound per class, acknowledging the epistemic uncertainty due to the availability of a limited amount of distributions. Such probability intervals over classes can be mapped on a convex set of probabilities (a credal set) from which, in turn, a unique prediction can be obtained using a transformation called *intersection probability transformation*. In this article, we conduct extensive experiments on several out-of-distribution (OOD) detection benchmarks, encompassing various dataset pairs (CIFAR10/100 vs SVHN/Tiny-ImageNet, CIFAR10 vs CIFAR10-C, CIFAR100 vs CIFAR100-C and ImageNet vs ImageNet-O) and using different network architectures (such as VGG16, ResNet-18/50, EfficientNet B2, and ViT Base). Compared to the BNN and DE baselines, the proposed credal wrapper method exhibits superior performance in uncertainty estimation and achieves a lower expected calibration error on corrupted data.

## 1 INTRODUCTION

Despite their success in various scientific and industrial areas, deep neural networks often generate inaccurate and overconfident predictions when faced with uncertainties induced by, e.g. out-of-distribution (OOD) samples, natural fluctuations, or adversarial disruptions (Papernot et al., 2016; Zhang et al., 2021; Hüllermeier & Waegeman, 2021; Hendrycks & Gimpel, 2016). Properly estimating the uncertainty associated with their predictions is key to improving the reliability and robustness of neural networks (Senge et al., 2014; Kendall & Gal, 2017; Sale et al., 2023b). Researchers commonly distinguish two types of uncertainty in neural networks (and machine learning models in general). Aleatoric uncertainty (AU), also known as data uncertainty, arises from inherent randomness such as data noise, and is irreducible. Epistemic uncertainty (EU) stems from the lack of knowledge of the data generation process and can be reduced with increased availability of training data (Hüllermeier & Waegeman, 2021). The distinction between AU and EU can be beneficial in applications such as active learning or OOD detection, particularly in safety-critical and practical fields such as autonomous driving (Zhou et al., 2012) and medical diagnosis (Lambrou et al., 2010). For example, in active learning, the objective is to avoid inputs that exhibit high AU but low EU. Similarly, effective EU estimation can prevent the misclassification of ambiguous in-distribution (ID) examples as OOD instances (Mukhoti et al., 2023), for the inherent ambiguity of the former does not arise from more epistemically uncertain regions of the ID distribution.

To represent and estimate both AU and EU, researchers have recently proposed neural networks able to output a 'second-order' representation in the target space, to express their uncertainty about a pre-

---

*Corresponding author.

diction's uncertainty itself (Hüllermeier et al., 2022; Sale et al., 2023b; Cuzzolin, 2024). Sampling-based approaches, among which Bayesian neural networks (BNNs) and deep ensembles (DEs), have emerged as powerful methods in this domain.

The BNN framework (Blundell et al., 2015; Gal & Ghahramani, 2016; Krueger et al., 2017; Mobiny et al., 2021), in particular, quantifies uncertainty by learning a posterior distribution over a network's weights; this, in turn, is used to predict a 'second-order' distribution (i.e., a distribution of distributions (Hüllermeier & Waegeman, 2021)) over the target space. While BNNs are straight-forward to construct, their training presents a significant challenge due to inherent computational complexity. Consequently, several approximation algorithms have been developed, such as sampling methods (Neal et al., 2011; Hoffman et al., 2014; Hobbhahn et al., 2022; Daxberger et al., 2021; Tran et al., 2020; Rudner et al., 2022), variational inference approaches (Blundell et al., 2015; Gal & Ghahramani, 2016), and Laplace approximation methodologies (Jospin et al., 2022). As computing the exact 'second-order' distribution generated by full Bayesian inference at prediction time is of prohibitive complexity, Bayesian model averaging (BMA) is often applied in practice (Gal & Ghahramani, 2016). The latter entails sampling a finite set of deterministic weight values from the posterior (obtained after training) to generate a collection of single (softmax) distributions. However, in practice, BNNs are generally challenging to scale to large datasets and architectures due to the high computational complexity (Mukhoti et al., 2023), and a limited number of samples is often used for BMA to reduce the inference complexity.

DEs (Lakshminarayanan et al., 2017), an established alternative approach for quantifying uncertainty, have been recently serving as a strong baseline in deep learning (Ovadia et al., 2019; Gustafsson et al., 2020; Abe et al., 2022; Mucsányi et al., 2024). DEs marginalize multiple standard neural network models to obtain a predictive distribution instead of explicitly inferring a distribution over parameters such as BNNs (Lakshminarayanan et al., 2017; Band et al., 2021). At prediction time, DEs average a finite set of single probability distributions for classification. Despite their successes in uncertainty quantification, a recent study has shown that DEs may demonstrate relatively low quality of EU estimation (Abe et al., 2022). They have also been criticized because of their substantial demand for memory and computational resources (Liu et al., 2020; He et al., 2020).

As previously discussed, a practical challenge in both BNNs and DEs at prediction time is the limited number of individual distributions (e.g., most papers use up to five) used to approximate the true predictive distribution and quantify uncertainty. Therefore, an intriguing research question arises here: *Can the uncertainty quantification performance of the sampling-based approaches be enhanced given the constrained number of predictive distributions?*

**Novelty and Contribution** In response, this paper presents an innovative method for formulating a credal set representation for both BNNs with BMA and DEs, capable of improving their uncertainty estimation in classification tasks, which we term *credal wrapper*. Given a limited collection of single distributions derived, at prediction time, from either BNNs or DEs, the proposed approach computes a lower and an upper probability bound per class, i.e., a collection of probability intervals (Tessem, 1992; De Campos et al., 1994), to acknowledge the epistemic uncertainty about the sought exact predictive distribution induced by the limited information actually available.

Such lower and upper bounds on each class' probability induce, in turn, a convex set of probabilities, known as *credal set* (Levi, 1980; De Campos et al., 1994). From such a credal set (the output of our credal wrapper), an *intersection probability* can be derived, in whose favor theoretical arguments exist as the natural way of mapping a credal set to a single probability (Cuzzolin, 2009; 2022).

Extensive experiments are performed on several OOD detection benchmarks including different dataset pairs (CIFAR10/100 vs SVHN/Tiny-ImageNet, CIFAR10 vs CIFAR10-C, CIFAR100 vs CIFAR100-C, and ImageNet vs ImageNet-O) and utilizing various network architectures (VGG16, ResNet-18/50, EfficientNet B2 and ViT Base). Compared to the BNN and DE baselines, the proposed *credal wrapper* demonstrates improved uncertainty estimation and a lower expected calibration error on the corrupted test instances when using the intersection probability for class prediction.

**Other Related Work** Numerous mathematical theories, such as subjective probability (De Finetti, 2017), possibility theory (Zadeh, 1978; Dubois & Prade, 1990), credal sets (Kyburg Jr, 1987; Levi, 1980), probability intervals, random sets (Nguyen, 1978), and imprecise probability theory (Walley, 1991), have been devised to estimate the (epistemic) uncertainty in response to the challenge posed by the limited availability of information (in our case, a limited number of sampled single

distributions). These theories state that the exact probability distribution is inaccessible and that the available evidence should instead be used to impose specific constraints on the unknown distribution.

Among them, probability intervals represent one of the simplest approaches and have attracted significant interest among researchers (Yager & Kreinovich, 1999; Guo & Tanaka, 2010; Wei et al., 2021; Cuzzolin, 2022). Instead of providing point-valued probabilities, probability intervals assume that the probability values $p(y)$ of the elements (e.g., classes) of the target space (e.g., the set of classes) belong to an interval $p_L(y) \leq p(y) \leq p_U(y)$, delimited by a lower probability bound $p_L(y)$ and an upper probability bound $p_U(y)$. Researchers have claimed that probability intervals may express uncertainty more appropriately than single probabilities (De Campos et al., 1994; Cano & Moral, 2002; Guo & Tanaka, 2010), particularly in situations where: (i) limited information is available to estimate unknown and exact probabilities; (ii) individual pieces of information are in conflict: e.g., when three predictors for the weather condition (rainy, sunny, or cloudy) predict probability vectors $(0.2, 0.6, 0.2)$, $(0.1, 0.2, 0.7)$, and $(0.7, 0.1, 0.2)$, respectively. Such undesirable scenarios can arise when BNNs or DEs use a limited number of predictive samples, as we discuss here.

Credal sets have recently attracted increasing attention within the broader field of machine learning for uncertainty quantification (Zaffalon, 2002; Corani & Zaffalon, 2008; Corani et al., 2012; Mauá et al., 2017; Hüllermeier & Waegeman, 2021; Shaker & Hüllermeier, 2021; Sale et al., 2023a). The advantage of adopting credal sets is the integration of notions of the set and probability distribution in a single and unified framework. Using credal sets, rather than individual distributions, arguably allows models to more naturally express epistemic uncertainty (Corani et al., 2012; Hüllermeier & Waegeman, 2021). Concerning deep neural networks, 'imprecise' BNNs (Caprio et al., 2024) modeling the network weights and predictions as credal sets have been proposed. Despite demonstrating robustness, the computational complexity of imprecise BNNs is comparable to that of the ensemble of BNNs, posing significant challenges for their widespread application. In addition, Wang et al. (2025) have introduced credal-set interval neural networks, which predict credal sets from probability intervals based on the deterministic interval output of interval neural networks. While effective in uncertainty quantification for small datasets, their scalability is limited by high computational costs.

In addition to BNNs and DEs, evidential deep learning models (Malinin & Gales, 2018; 2019; Malinin et al., 2019; Charpentier et al., 2020) have also been developed that generate a Dirichlet distribution (another type of 'second-order' representation) in the target space. A drawback of these methods is the fact that suitable Dirichlet distribution labels are not available at training time. Further, the behaviour of these models often deviates from theoretical EU assumptions (Ulmer et al., 2023). Recent research (Juergens et al., 2024) has shown that evidential methods often fail to faithfully represent EU, and that the resulting EU measures lack a meaningful quantitative interpretation.

**Paper Outline** The rest of the paper is structured as follows. Sec. 2 presents how uncertainty estimation works in different model classes. Sec. 3 introduces our credal wrapper in full detail. Sec. 4 describes the experimental validations. Sec. 5 summarizes our conclusion and future work. Appendices report additional experiments in §A and implementation details in §B, respectively.

## 2 Uncertainty Estimation in Different Model Classes

**Bayesian Neural Networks** BNNs model network parameters, i.e., weights and biases, as probability distributions. The resulting predictive distributions can thus be seen as 'second-order' distributions, i.e., probability distributions of distributions (Hüllermeier & Waegeman, 2021). As mentioned, for computational reasons, BMA is often applied for BNN inference (Gal & Ghahramani, 2016). Namely, in a classification context, we obtain

$$\tilde{\boldsymbol{p}} = \frac{1}{N_p} \sum_{i=1}^{N_p} \boldsymbol{p}_i = \frac{1}{N_p} \sum_{i=1}^{N_p} h_{\text{bnn}}^{\boldsymbol{\omega}_i}(\boldsymbol{x}), \tag{1}$$

where $\boldsymbol{p}_i$ is the sampled prediction from the deterministic model ($h_{\text{bnn}}^{\boldsymbol{\omega}_i}$) parametrized by $\boldsymbol{\omega}_i$, $N_p$ is the number of samples, and $\tilde{\boldsymbol{p}}$ is the averaged probability. The parameter vector $\boldsymbol{\omega}_i$ is obtained from the $i$-th sampling instance of the parameter posterior distribution of BNNs (Jospin et al., 2022).

Employing Shannon entropy as the uncertainty measure, the total uncertainty (TU) and AU in BNNs can be approximately quantified by calculating the entropy of the averaged prediction and averaging the entropy of each sampled prediction (Hüllermeier & Waegeman, 2021), respectively, as follows:

$$\text{TU} := H(\tilde{\boldsymbol{p}}) = -\sum_k^C \tilde{q}_k \log_2 \tilde{q}_k, \ \text{AU} := \tilde{H}(\boldsymbol{p}) = \frac{1}{N_p}\sum_{i=1}^{N_p} H(\boldsymbol{p}_i) = -\frac{1}{N_p}\sum_{i=1}^{N_p}\sum_k^C p_{i_k}\log_2 p_{i_k}, \tag{2}$$

where $\tilde{q}_k$ and $p_{i_k}$ are the $k$-th elements of the probability vectors $\tilde{\boldsymbol{p}}$ and $\boldsymbol{p}_i$ across the $C$ classes, respectively. An estimate of epistemic uncertainty can then be derived as EU $:= H(\tilde{\boldsymbol{p}}) - \tilde{H}(\boldsymbol{p})$ (Depeweg et al., 2018), which can be interpreted as an approximation of 'mutual information' (Hüllermeier et al., 2022; Hüllermeier & Waegeman, 2021).

**Deep Ensembles** DEs generate an averaged prediction from a set of $M$ individually-trained standard neural networks (SNNs), as follows:

$$\tilde{\boldsymbol{p}} = \tfrac{1}{M}\sum_{m=1}^M h_m(\boldsymbol{x}) = \tfrac{1}{M}\sum_m^M \boldsymbol{p}_m, \tag{3}$$

where $\boldsymbol{p}_m$ denotes the single probability vector provided by the $m$-th SNN. Considered by some to be an approximation of BMA (Wilson & Izmailov, 2020; Abe et al., 2022), DEs also quantify TU, AU, and EU as in eq. (2), where the sampled single probability vector $\boldsymbol{p}_i$ is replaced by $\boldsymbol{p}_m$.

**Credal Sets** Uncertainty quantification for credal sets is an active research subject (Hüllermeier & Waegeman, 2021; Sale et al., 2023a). To that extent, researchers have developed the concepts of generalized entropy (Abellán et al., 2006) and generalized Hartley (GH) measure (Abellán & Moral, 2000; Hüllermeier et al., 2022). However, applying the GH measure in practice is challenging due to the high computational cost of solving constrained optimization problems involving $2^C$ subsets (Hüllermeier & Waegeman, 2021; Hüllermeier et al., 2022), particularly when the number of classes $C$ is large (e.g., $C = 100$). The width of the probability interval has also been proposed as an EU measure (Hüllermeier et al., 2022), but is only applicable to binary cases. As a result, we opt for generalized entropy. Given a credal set prediction, denoted by $\mathbb{P}$, its upper and lower entropy, $\overline{H}(\mathbb{P})$ and $\underline{H}(\mathbb{P})$, can be calculated as a generalization of the classical Shannon entropy, allowing us to estimate the TU and AU for a credal prediction (Abellán et al., 2006), as follows:

$$\text{TU} := \overline{H}(\mathbb{P}) = \max_{\boldsymbol{p}\in\mathbb{P}} H(\boldsymbol{p}), \quad \text{AU} := \underline{H}(\mathbb{P}) = \min_{\boldsymbol{p}\in\mathbb{P}} H(\boldsymbol{p}). \tag{4}$$

Such measures capture the maximal and the minimal Shannon entropy within the credal set, respectively. The level of EU can then be measured by their difference, namely EU $:= \overline{H}(\mathbb{P}) - \underline{H}(\mathbb{P})$.

## 3 METHODOLOGY

Inspired by the use of probability intervals for decision-making (Yager & Kreinovich, 1999; Guo & Tanaka, 2010), we propose to build probability intervals by extracting the upper and lower bound per class from the given set of limited (categorical) probability distributions, validating this choice via extensive experiments in Sec. 4. E.g., consider again the task of predicting weather conditions (rainy, sunny, or cloudy). When receiving three probability values for the *rainy* condition, e.g., 0.2, 0.1, and 0.7, using probability intervals we model the uncertainty on the probability of the rainy condition as $[0.1, 0.7]$. Each probability interval system can determine a convex set of probabilities over the set of classes, i.e., a credal set. Such a credal set is a more natural model than individual distributions for representing the epistemic uncertainty encoded by the prediction, as it amounts to constraints on the unknown exact distribution (Hüllermeier & Waegeman, 2021; Shaker & Hüllermeier, 2021; Sale et al., 2023a). Nevertheless, a single predictive distribution, termed *intersection probability*, can still be derived from a credal set to generate a unique class prediction for classification purposes. Our credal wrapper framework is depicted in Figure 1. The remainder of this section discusses the credal wrapper generation, a method for computational complexity reduction for uncertainty estimation, and the intersection probability, in this order.

**Credal Wrapper Generation** Given a set of $N$ individual distributions from BNNs or DEs, an upper and a lower probability bound for $k$-th class element, denoted as $p_{U_k}$ and $p_{L_k}$, respectively, can be obtained from

$$p_{U_k} = \max_{n=1,..,N} p_{n,k}, \quad p_{L_k} = \min_{n=1,..,N} p_{n,k}, \tag{5}$$

where $p_{n,k}$ denotes the $k$-th element of the $n$-th single probability vector $\boldsymbol{p}_n$. Such probability intervals over $C$ classes determine a non-empty credal set $\mathbb{P}$, as follows (Moral-García & Abellán, 2021; De Campos et al., 1994):

$$\mathbb{P} = \{\boldsymbol{p} \mid \boldsymbol{p} \in \Delta^{C-1} \text{ and } p_k \in [p_{L_k}, p_{U_k}] \; \forall k = 1, 2, ..., C\} \text{ s.t. } \sum_k^C p_{L_k} \leq 1 \leq \sum_k^C p_{U_k}, \tag{6}$$

where $\boldsymbol{p}$ denotes the valid (normalized) probability vector and $\Delta^{C-1}$ represents the probability simplex on the set of $C$ classes. The condition in eq. (6) guarantees that individual probabilities in

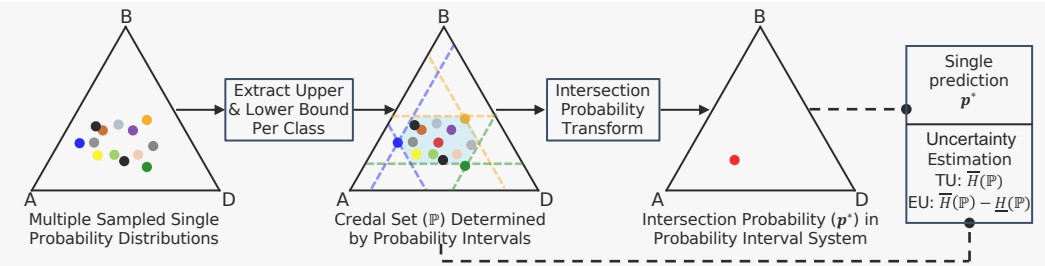

Figure 1: Credal wrapper framework for a three-class (A, B, D) classification task. Given a set of individual probability distributions (denoted as single dots) in the simplex (triangle) of probability distributions of the classes, probability intervals (parallel lines) are derived by extracting the upper and lower probability bounds per class, using eq. (5). Such lower and upper probability intervals induce a credal set on $\{A, B, D\}$ ($\mathbb{P}$, light blue convex hull in the triangle). A single intersection probability (the red dot) is computed from the credal set using the transform in eq. (5). Uncertainty is estimated in the mathematical framework of credal sets in eq. (4).

the credal set satisfy the normalization condition, and their probability value per class is constrained to the given probability interval. It can be readily proven that the probability intervals generated in eq. (5) meet the above condition, as follows:

$$\sum_k^C p_{L_k} = \sum_k^C \min_{n=1,..,N} p_{n,k} \leq \sum_k^C p_{n^*,k} = 1 \leq \sum_k^C \max_{n=1,..,N} p_{n,k} = \sum_k^C p_{U_k}, \qquad (7)$$

where $n^*$ is any index in $1, \ldots, N$. Computing $\overline{H}(\mathbb{P})$ in eq. (4) for uncertainty estimation, on the other hand, requires solving the following optimization problems:

$$\overline{H}(\mathbb{P}) = \text{maximize} \sum_k^C -p_k \cdot \log_2 p_k \text{ s.t. } \sum_k^C p_k = 1 \text{ and } p_k \in [p_{L_k}, p_{U_k}] \,\forall k = 1, ..., C, \qquad (8)$$

which can be addressed by using a standard solver, such as the SciPy optimization package (Virtanen et al., 2020). To calculate $\underline{H}(\mathbb{P})$ in eq. (4), maximize is simply replaced by minimize. We further discuss an alternative approach for generating credal sets in Appendix §C.

**Computational Complexity Reduction Method** The convex optimization problem in eq. (8) may present a computational challenge for a large value of $C$ (e.g., $C = 1000$). To mitigate this issue, we propose an approach, termed *Probability Interval Approximation* (PIA), in Algorithm 1. The PIA method initially identifies the top $J - 1$ relevant classes by sorting the probability values of the intersection probability $\boldsymbol{p}^*$ (detailed in eq. (9)) below) in descending order. Then, the remaining elements are merged into a single class whose upper and lower probability are computed. As a result, the dimension of the approximate probability interval is reduced from $C$ to $J$.

---

**Algorithm 1** Probability Interval Approximation

**Input:** $[p_{L_k}, p_{U_k}]$ for $k = 1, ..C$; Intersection probability $\boldsymbol{p}^*$; Chosen number of classes $J$
**Output:** Approximated probability intervals $[r_{L_j}, r_{U_j}]$ for $k = 1, ..J$
 1: Compute the index vector for sorting $\boldsymbol{p}^*$ in descending order as
    $\boldsymbol{m} := (m_1, m_2, ..., m_C)$ for $p^*_{m_1} \geq p^*_{m_2} \geq ... \geq p^*_{m_C}$
 2: Calculate the upper and lower probability per selected class as
    $r_{L_j} := p_{L_{m_j}}, r_{U_j} := p_{U_{m_j}}$ for $j = 1, ..., J-1$
 3: Calculate upper and lower probability for deselected classes as
    $r_{L_J} = \max(1 - \sum_{i=m_J}^{m_C} p_{U_i}, \sum_{j=1}^{J-1} r_{L_j}); r_{U_J} = \min(1 - \sum_{i=m_J}^{m_C} p_{L_i}, \sum_{j=1}^{J-1} r_{U_j})$

---

**Intersection Probability** In classification tasks, it is desirable to eventually map a credal set to a single probability prediction to make a unique class prediction. In our credal wrapper, from the probability interval predictions defined in eq. (5), we derive a single *intersection probability* vector for class prediction (Cuzzolin, 2022) as the unique probability distribution which (i) ensures that the derived single probability is normalized within the probability interval system, and (ii) assumes equal trust in the probability intervals for each class, since there is no justification for treating them differently (Cuzzolin, 2022). Mathematically, the intersection probability, $\boldsymbol{p}^*$, must satisfy

$$p_k^* = p_{L_k} + \alpha \cdot (p_{U_k} - p_{L_k}) \,\forall k = 1, ..., C \qquad (9)$$

where $p_k^*$ is the $k$-th element of $\boldsymbol{p}^*$ under a constant $\alpha \in [0, 1]$. This is a convex combination of $p_{L_k}$ and $p_{U_k}$, so that $p_k^* \in [p_{L_k}, p_{U_k}]$. Due to the normalization constraint, $\sum_k^C p_k^* = 1$, it follows that:

$$\sum_k^C \big(p_{L_k} + \alpha(p_{U_k} - p_{L_k})\big) = 1 \Rightarrow \sum_k^C (p_{L_k}) + \alpha \sum_k^C (p_{U_k} - p_{L_k}) = 1. \tag{10}$$

As a result, the unique $\alpha$ can be computed as follows (Cuzzolin, 2022):

$$\alpha = \big(1 - \sum_k^C p_{L_k}\big) / \big(\sum_k^C (p_{U_k} - p_{L_k})\big). \tag{11}$$

The intersection probability is thus given by eq. (9) with $\alpha$ as given in eq. (11). The intersection probability transform follows clear rationale principles and does not suffer from the drawbacks of other potential transforms. Let us see this in detail. (i) Taking the average probability value of the original sampled distributions produces a valid probability vector that falls within the probability interval system. However, such a transform does not leverage the probability bounds, thus violating the semantics of probability intervals (Guo & Tanaka, 2010; Cano & Moral, 2002) (and also exhibiting inferior performance in the experiments). (ii) Selecting the midpoint of each probability interval $[p_L, p_U]$, instead, does not generally result in a valid normalized probability vector. (iii) Normalizing the lower or the upper probability vectors, i.e., taking $\hat{p}_{U_k} = p_{U_k} / \sum_k^C p_{U_k}$ or $\hat{p}_{L_k} = p_{L_k} / \sum_k^C p_{L_k}$, yields a probability that is not guaranteed to be consistent with the interval system (Cuzzolin, 2020). In Appendix §D, we further discuss the limit behavior of the intersection probability when the number of sampled probability distributions goes to infinity.

## 4  EXPERIMENTAL VALIDATION

In this section, we apply OOD detection benchmarks for EU quantification assessment OOD (Mukhoti et al., 2023; Mucsányi et al., 2024). Regarding OOD detection, a model is expected to exhibit high EU estimates on OOD samples in comparison to in-domain (ID) instances. Consequently, superior OOD detection performance provides evidence of the enhanced uncertainty estimation quality. To assess OOD detection performance, we use the AUROC (Area Under the Receiver Operating Characteristic curve) and AUPRC (Area Under the Precision-Recall curve) scores. Greater scores indicate a higher quality of uncertainty estimation. In addition to OOD detection performance, expected calibration error (ECE) (Guo et al., 2017; Nixon et al., 2019) is also applied to measure the model uncertainty and calibration performance. A lower ECE value signifies a closer alignment between the model's confidence scores and the true probabilities of the events.

**Evaluation using Small-scale Datasets**  In this experiment, we use dataset pairs of the type (ID samples vs OOD data), including CIFAR10 (Krizhevsky et al., 2009) vs SVHN (Hendrycks et al., 2021)/Tiny-ImageNet (Le & Yang, 2015) and CIFAR10 vs CIFAR10-C (Hendrycks & Dietterich, 2019). As for BNN baselines, we choose two standardized variational BNNs: BNNR (Auto-Encoding variational Bayes (Kingma & Welling, 2013) with the local reparameterization trick (Molchanov et al., 2017)) and BNNF (Flipout gradient estimator with negative evidence of lower bound loss (Wen et al., 2018)). We do not consider BNNs using sampling approaches because of their generally high computational resource requirements (Gawlikowski et al., 2021; Jospin et al., 2022). As for DEs, we aggregate five standard neural networks, trained using distinct random seeds. All models are implemented on the established VGG16 (Simonyan & Zisserman, 2015) and ResNet-18 (He et al., 2016) architectures using the CIFAR10 dataset. The number of sampled predictions for BNNs is set to $N = 5$. While the main aim of this work is to improve the uncertainty estimation performance of BNNs and DEs through the credal wrapper, it is also intriguing to compare our approach with an alternative method, Ensemble Distribution Distillation (EDD) (Malinin et al., 2019), within the evidential deep learning family. EDD aims to approximate DE predictions on the simplex by leveraging a single Dirichlet distribution. Thus, here we train EDD from the DEs under two distinct learning rate scheduler configurations: (1) EDD-Fair, which uses the same basic scheduler as BNNs and SNNs to ensure a fair comparison; and (2) EDD, which employs the cycle learning rate policy described in the original EDD paper. More implementation details are given in Appendix §B.

Table 1 reports test accuracy (ACC) and ECE values on the CIFAR10 test set and the corrupted CIFAR10 (CIFAR10-C) data as well as the OOD detection performance comparison. Regarding the CIFAR10-C dataset, which applies 15 corruption modes to the CIFAR10 instances, with 5 intensities per corruption type, Figures 2 and 3 show the OOD detection results and the ECE values of CIFAR10 vs CIFAR10-C against increased corruption intensity, respectively. The results over the 15

Table 1: Performance comparison between the classical and credal wrapper version of BNN and DE, as well as EDD models. All models are implemented on VGG16/ResNet-18 backbones and trained using CIFAR10 data as ID samples. The results are from 15 runs. The best scores per metric are in bold. The results on corrupted data are averaged over all corruption types and intensities.

| | | | Prediction Performance | | | | OOD Detection Performance (%) using EU | | | |
| --- | --- | --- | --- | --- | --- | --- | --- | --- | --- | --- |
| | | | ID Test Data | | Corrupted Data | | SVHN (OOD) | | Tiny-ImageNet (OOD) | |
| | | | ACC (%) | ECE | ACC (%) | ECE | AUROC | AUPRC | AUROC | AUPRC |
| VGG16 | BNNR | Baseline | **90.98±0.11** | 0.048±0.003 | 21.86±1.88 | 0.470±0.050 | 86.65±1.26 | 90.61±0.88 | 84.62±0.28 | 80.06±0.40 |
| | | Ours | **90.98±0.15** | **0.034±0.003** | **21.96±1.83** | **0.416±0.050** | **87.85±1.45** | **92.32±1.05** | **85.55±0.30** | **82.76±0.46** |
| | BNNF | Baseline | **90.88±0.19** | 0.055±0.003 | 23.00±1.97 | 0.498±0.066 | 86.79±0.47 | 90.76±0.55 | 84.54±0.20 | 79.91±0.35 |
| | | Ours | 90.84±0.20 | **0.043±0.003** | **23.08±1.93** | **0.452±0.068** | **87.66±0.50** | **92.26±0.51** | **85.26±0.20** | **82.33±0.30** |
| | DE | Baseline | **93.57±0.09** | **0.015±0.001** | **24.81±0.79** | 0.314±0.024 | 89.74±1.31 | 93.58±0.97 | 88.49±0.17 | 85.79±0.35 |
| | | Ours | 93.47±0.10 | 0.033±0.002 | 24.42±0.81 | 0.200±0.022 | **91.96±1.33** | **95.24±0.79** | **89.80±0.08** | **87.99±0.16** |
| | | EDD | 91.05±0.18 | 0.038±0.003 | 19.37±2.24 | **0.122±0.062** | 91.21±0.95 | 93.62±1.70 | 87.04±0.75 | 82.53±2.18 |
| ResNet-18 | BNNR | Baseline | **91.87±0.17** | 0.058±0.002 | 22.38±1.31 | 0.498±0.030 | 88.32±1.22 | 93.03±0.74 | 85.83±0.25 | 81.43±0.51 |
| | | Ours | 91.86±0.18 | **0.055±0.002** | **22.45±1.30** | **0.460±0.030** | **88.40±1.25** | **93.16±0.79** | **85.99±0.20** | **82.51±0.39** |
| | BNNF | Baseline | **91.91±0.19** | 0.057±0.002 | 22.38±1.30 | 0.498±0.024 | 88.62±0.80 | 93.26±0.52 | 85.94±0.30 | 81.64±0.34 |
| | | Ours | 91.90±0.18 | **0.055±0.002** | **22.43±1.30** | **0.464±0.024** | **88.71±0.81** | **93.41±0.51** | **86.06±0.31** | **82.69±0.27** |
| | DE | Baseline | **93.36±0.11** | **0.018±0.001** | 26.62±0.69 | 0.304±0.018 | 87.40±0.68 | 91.82±0.51 | 87.94±0.15 | 84.86±0.24 |
| | | Ours | 93.35±0.13 | 0.025±0.002 | 26.41±0.73 | **0.224±0.016** | **88.63±0.80** | **92.76±0.64** | **89.16±0.15** | **87.41±0.19** |
| | | EDD | 92.96±0.10 | 0.027±0.001 | **27.00±1.20** | 0.276±0.032 | 87.26±1.57 | 91.91±1.01 | 88.05±0.23 | 83.66±0.41 |

corruption types are averaged. Table 1 and Figure 3 demonstrate that our intersection probability exhibits comparable test ACC and ECE on ID test data, compared to the classical averaged probability. Moreover, the calibration performance on corrupted samples is consistently better, as evidenced by the lower ECE values. Table 1 and Figure 2 also demonstrate that our credal wrapper markedly and persistently enhances the EU estimation of baselines. This is evidenced by the augmented OOD detection performance on various data pairs and across disparate backbones. Compared to EDD, which applies a cycle learning rate policy, our credal wrapper still consistently achieves superior performance, as shown in Table 1 and Figure 2. Concerning EDD-Fair, we observe that fairly applying the simple learning rate scheduler leads to low prediction performance of EDD-Fair (considerably below the baselines) and poor EU estimation (evidenced by the lowest OOD detection values), as shown in Table 19 in the Appendix.

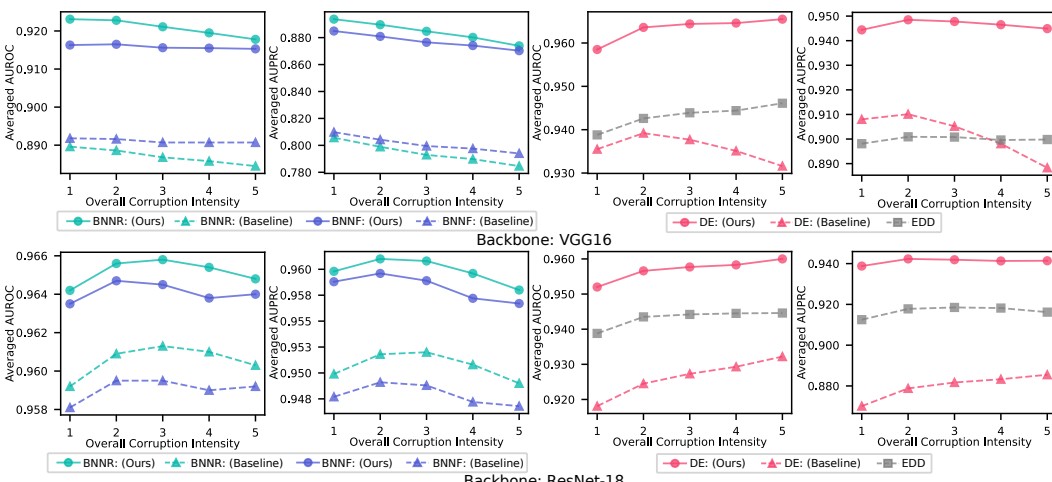

Figure 2: OOD detection using EU as the metric on CIFAR10 vs CIFAR10-C of the classical and credal wrapper version of BNNs and DE, and EDD against increased corruption intensity, using VGG16 and ResNet-18 as backbones.

**Evaluation using Large-scale Datasets** In this experiment, we exclusively employ DEs (combining five SNNs) to assess the quality of EU estimation of our credal wrapper on large-scale datasets and network architectures. This is because, in practice, BNNs are typically unable to scale to large datasets and model architectures due to the high computational complexity (Mukhoti et al., 2023).

Figure 3: ECE values of BNNR, BNNF, and DE on CIFAR10-C against increased corruption intensity, using the averaged probability (Prob.) and our proposed intersection probability (Prob.). VGG16 and ResNet-18 are backbones. Results are from 15 runs.

Table 2: OOD detection AUROC and AUPRC performance (%) of both the classical and credal wrapper version of DEs using EU as the metric. The results are from 15 runs, based on the ResNet-50 backbone. Best scores are in bold.

| | CIFAR10 (ID) | | | | CIFAR100 (ID) | | | | ImageNet (ID) | |
|---|---|---|---|---|---|---|---|---|---|---|
| | SVHN (OOD) | | Tiny-ImageNet (OOD) | | SVHN (OOD) | | Tiny-ImageNet (OOD) | | ImageNet-O (OOD) | |
| | AUROC | AUPRC | AUROC | AUPRC | AUROC | AUPRC | AUROC | AUPRC | AUROC | AUPRC |
| Baseline | 89.58±0.93 | 92.29±1.00 | 86.87±0.20 | 83.02±0.16 | 73.83±1.97 | 84.96±1.25 | 78.80±0.20 | 74.68±0.27 | 65.70±0.41 | 63.20±0.35 |
| Ours | **93.77±0.60** | **96.06±0.46** | **88.78±0.15** | **86.83±0.23** | **80.22±1.96** | **89.40±1.03** | **81.00±0.16** | **77.16±0.23** | **66.20±0.38** | **63.23±0.34** |

For instance, training a ResNet-50-based BNN on CIFAR-10 (resized to (224, 224, 3)) failed in our experiment due to exceeding the memory capacity of a single Nvidia A100 GPU. The dataset pairs (ID vs OOD) considered include CIFAR10/CIFAR100 (Krizhevsky, 2012) vs SVHN/Tiny-ImageNet, ImageNet (Deng et al., 2009) vs ImageNet-O (Hendrycks et al., 2021), CIFAR10 vs CIFAR10-C, and CIFAR100 vs CIFAR100-C (Hendrycks & Dietterich, 2019). DEs are implemented on the well-established ResNet-50 (He et al., 2016). All input data have a shape of (224, 224, 3). More training details are given in Appendix §B. The PIA algorithm (Algorithm 1) is applied using the settings $J = 20$ and $J = 50$ to calculate the generalized entropy ($\overline{H}(\mathbb{P})$ and $\underline{H}(\mathbb{P})$) on dataset pairs involving CIFAR100 and ImageNet, respectively. Compared to classical DEs, our credal wrapper demonstrates the enhanced OOD detection across a spectrum of data pairs, as shown in Table 2, suggesting that our proposed method can consistently improve EU estimation.

**Ablation Study on Network Architectures** We also perform an ablation study using Efficient-NetB2 (EffB2) (Tan & Le, 2021) and Vision Transformer Base (ViT-B) (Dosovitskiy et al., 2020) as architecture backbones, involving CIFAR10/100 vs SVHN and Tiny-ImageNet, CIFAR10 vs CIFAR10-C, and CIFAR100 vs CIFAR100-C. The key findings, presented in Table 3 and Figure 4, demonstrate that our credal wrapper approach significantly enhances the EU quality of DEs, and is robust against both the dataset pairs and the architecture backbones. Furthermore, Figure 5 shows the ECE values exhibited by DEs on both CIFAR10-C and CIFAR100-C datasets, using ResNet-50, EffB2, and ViT-B backbones. The results indicate that adopting the intersection probability as the final prediction, with its strong rationality foundations, can indeed empirically improve calibration performance (lower ECE) on the corrupted samples by using the classical averaged probability.

Table 3: OOD detection AUROC and AUPRC performance (%) of the classical and credal wrapper version of DEs using EU as the metric. Results are from 15 runs, based on EffB2 and ViT-B backbones. Best scores are in bold.

| | | CIFAR 10 (ID) | | | | CIFAR 100 (ID) | | | |
|---|---|---|---|---|---|---|---|---|---|
| | | SVHN (OOD) | | Tiny-ImageNet (OOD) | | SVHN (OOD) | | Tiny-ImageNet (OOD) | |
| | | AUROC | AUPRC | AUROC | AUPRC | AUROC | AUPRC | AUROC | AUPRC |
| EffB2 | Baseline | 95.76±0.59 | 97.43±0.47 | 92.32±0.14 | 90.72±0.22 | 87.52±1.52 | 93.81±0.80 | 85.29±0.15 | 82.98±0.24 |
| | Ours | **97.60±0.25** | **98.74±0.14** | **93.55±0.15** | **93.17±0.16** | **89.20±1.19** | **94.58±0.56** | **86.23±0.10** | **83.69±0.21** |
| ViT-B | Baseline | 77.71±1.67 | 85.82±1.14 | 82.27±0.79 | 78.85±0.81 | 81.41±1.33 | 89.00±0.96 | 81.18±0.31 | 77.46±0.49 |
| | Ours | **80.71±1.83** | **88.43±1.26** | **84.28±0.76** | **82.82±0.85** | **84.87±1.06** | **91.40±0.67** | **82.53±0.32** | **79.04±0.39** |

**Ablation Study on Hyperparameter of PIA Algorithm** In this experiment, we investigate the effect of the hyperparameter $J$ of the PIA algorithm 1. The OOD detection tasks include CIFAR100 (ID) vs SVHN (OOD) and Tiny-ImageNet (OOD) and deep ensembles are implemented on the ResNet-50 backbone. Table 4 shows OOD detection performance and the time cost for uncertainty quantification per sample, using different settings of $J$ for the PIA algorithm.

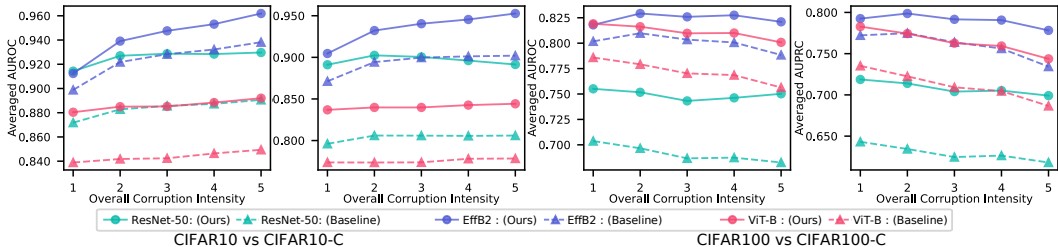

Figure 4: OOD detection performance of the classical and credal wrapper version of DEs using EU as the metric on CIFAR10/100 vs CIFAR10-C/100-C against increased corruption intensity, using ResNet-50, EffB2, and ViT-B as backbones.

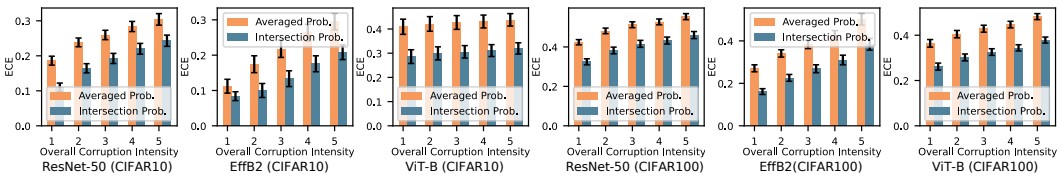

Figure 5: ECE values of DEs on CIFAR10-C and CIFAR100-C against increased corruption intensity, using the averaged probability (Prob.) and our proposed intersection probability (Prob.). ResNet-50, EffB2, and ViT-B are backbones. Results are from 15 runs.

The hyperparameter $J$ is a design choice intended to mitigate the computational cost associated with uncertainty quantification, particularly in situations with limited computational resources. The evaluation indicates that the proposed PIA algorithm can significantly reduce the computational complexity of estimating uncertainty in cases of high-class classification while maintaining superior uncertainty performance on OOD detection benchmarks. We can also observe that increasing the value of $J$ improves the OOD detection performance, but can lead to an increase in execution time. This is because solving the constrained optimization problem in eq. (8) involves more variables and constraints. Furthermore, we present the uncertainty estimates for ID and OOD samples at different values of $J$ in Figure 16 in Appendix §A.10. The result shows that using a small value of $J$ leads to an underestimation of uncertainty values, particularly for OOD samples. This phenomenon explains why a higher value of $J$ can achieve better OOD detection performance.

Table 4: OOD detection AUROC and AUPRC performance (%) of credal wrapper of DEs using EU as uncertainty metrics and the time cost, using different setting of $J$ of PIA algorithm. The OOD detection involves CIFAR100 (ID) vs SVHN (OOD) and Tiny-ImageNet (OOD). The results are from 15 runs, based on the ResNet-50 backbone.

| $J$ | EU as Metric | | | | Time Cost per Sample (ms) |
| | SVHN | | Tiny-ImageNet | | |
| | AUROC | AUPRC | AUROC | AUPRC | |
| --- | --- | --- | --- | --- | --- |
| 10 | 78.81±1.87 | 87.96±1.09 | 80.17±0.19 | 75.25±0.26 | 7.182±0.087 |
| 20 | 80.22±1.96 | 89.40±1.03 | 81.00±0.16 | 77.16±0.23 | 11.346±0.038 |
| 100 | 80.55±1.99 | 89.68±1.04 | 81.16±0.16 | 77.76±0.21 | 109.469±1.330 |

**Computational Cost** The reported time in Table 4 cost is measured on a single Intel Xeon Gold 8358 CPU@2.6 GHz, without optimization in the calculation process. As a reference, the time cost of the classical method for uncertainty quantification per sample is $0.007_{\pm0.001}$ms. In addition, the time cost for CIFAR10 without reduction can also be seen as 7.18ms from Table 4. While a bit high, it remains practical without limiting computational resources. We also believe a more efficient code implementation of our approach could significantly mitigate the computational cost.

Moreover, our method can achieve superior uncertainty quantification performance with fewer predictive samples, as demonstrated in Tables 5 and 6 in the Appendix. Additionally, the computational cost analysis in Appendix §A.9 indicates that the overhead introduced by our method for entropy calculation is negligible, particularly when accounting for the significant reduction in overall inference cost. From this perspective, the proposed credal wrapper not only reduces computational complexity but also enhances practical applicability.

**Additional Experiments** Appendix §A.1 presents visualizations of the credal set predictions and intersection probability, offering an intuitive graphical explanation of why our credal wrapper outperforms the baseline methods. Appendix §A.2 and §A.3 present ablation studies on the number of predictive samples in DEs and BNNs, respectively. The results consistently demonstrate the superior performance of our credal wrapper method. Appendix §A.4 examines the EU quantification performance of our credal wrapper using the generalized Hartley measure (Abellán & Moral, 2000), as an alternative to the difference between upper and lower entropy in eq. (4). The results demonstrate that our credal wrapper consistently enhances the EU estimation performance while being agnostic to the specific EU measure for credal sets employed. Appendix §A.5 assesses the TU estimation quality (as opposed to EU) on OOD detection benchmarks, suggesting that our credal wrapper consistently improves TU estimation as well, across the different dataset pairs, network architectures, and test settings. Appendix §A.6 presents an examination of the performance of our credal wrapper in scenarios where the ensemble member (SNN) generates overconfident predictions. The findings also show the improved uncertainty quantification of our credal wrapper. Appendix §A.7 presents uncertainty estimate plots for both ID and OOD samples across various settings, providing further qualitative evidence that our credal wrapper consistently enhances baseline uncertainty quantification. Appendix §A.8 assesses intersection probability on corrupted samples using the negative log-likelihood (NLL) metric, showing consistently improved performance with lower values across extensive test scenarios.

## 5    CONCLUSION AND FUTURE WORK

**Conclusion** This paper presents an innovative approach, called a *credal wrapper*, to formulating a credal set prediction for model averaging in BNNs and DEs, capable of improving their EU and TU estimation in classification tasks. Given a limited collection of single distributions derived at prediction time from either BNNs or DEs, the proposed approach computes a lower and an upper probability bound per class, to acknowledge the epistemic uncertainty about the sought exact predictive distribution due to limited information availability. Such lower and upper probability bounds on each class induce a credal set. From such a credal prediction (the output of our wrapper), an intersection probability distribution can be derived, for which there are theoretical arguments that it is the natural way to map a credal set to a single probability.

Extensive experiments performed on several OOD detection benchmarks, including CIFAR10/100 vs SVHN/Tiny-ImageNet, CIFAR10/100 vs CIFAR10/100-C, and ImageNet vs ImageNet-O, and utilizing various network architectures (VGG16, ResNet-18/50, EfficientNet B2 and ViT Base), demonstrate that, compared to BNN and DE baselines, the proposed credal wrapper approach exhibits better uncertainty estimation and lower ECE on corrupted data.

**Limitation and Future Work** Our credal wrapper functions as a plug-and-play method for enhancing the uncertainty quantification of BNNs and DEs without requiring any additional training. However, uncertainty quantification in this setting demands a higher computational investment. Thus, if memory usage or computational resources are strictly limited, the proposed approach may not be the optimal solution. A primary objective of our forthcoming research is to apply and benchmark our approach on safety-critical and real-world applications, such as medical image analysis.

## ACKNOWLEDGMENT

We thank the anonymous reviewers for their valuable feedback. This work has received funding from the European Union's Horizon 2020 research and innovation program under grant agreement No. 964505 (E-pi).

## REPRODUCIBILITY STATEMENT

Detailed implementation description is provided in §B. The experiment codes are provided here.

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

TABLE OF APPENDIX CONTENT

## A   ADDITIONAL EXPERIMENTS

### A.1   VISUALIZATION OF CREDAL SET PREDICTIONS AND INTERSECTION PROBABILITY

Visualizing credal sets in high-dimensional ($C$) classification tasks is challenging, as it requires a $(C-1)$-dimensional probability simplex. To demonstrate our credal wrapper method, we consider a three-class classification scenario derived from the CIFAR10 classification tasks in the main body, focusing on the classes 'airplane', 'automobile', and 'others'. The class 'others' includes the remaining class indices in the original CIFAR10 datasets. We visualize our credal wrapper of ResNet50-based DEs and VGG16-based BNNRs using different sample sizes on the 2D simplex in Figures 6 and 7. Three cases are considered: the correctly classified in-distribution (ID) sample (left), the incorrectly classified sample (middle), and the OOD samples (right).

From the visualizations, we observe that the differences between individual predictive probability samples for correctly classified ID data and the resulting discrepancy between the intersection probability and the averaged probability are less visible. However, for incorrectly classified samples, the differences become more apparent. Since the intersection probability is derived from probability intervals, it tends to exhibit less overconfidence in incorrect predictions. A similar observation can be seen in Figure 8, where the individual probabilities, the corresponding intersection probability, and the averaged probability for the CIFAR10 and CIFAR10-C (corrupted) samples are visualized. Furthermore, as the ensemble size increases, this reduction in overconfidence becomes even more apparent (Figures 6 and 7). These observations explain the comparable performance of the intersection probability and the averaged probability in terms of test accuracy and expected calibration error (ECE) on ID data but improved performance on corrupted data.

Figures 6 and 7 also illustrate that the credal sets for incorrectly classified and OOD samples are notably larger than those for correctly classified ID samples. This behavior aligns with the construction of our credal wrapper, which is formulated by extracting the upper and lower probability limits per class from individual probability samples. It also shows that the credal sets naturally expand when newly sampled probabilities fall outside the original bounds (for example, the OOD cases in Figures 6 and 7), reflecting a conservation property that reasonably incorporates new evidence rather than disregarding it. This phenomenon can also explain why the OOD detection performance can

be further enhanced with an increased sample size, as observed in Tables 5 and 6. Furthermore, the theoretical foundation for uncertainty quantification using credal sets and probability interval systems is robust, as detailed in Appendix §A.4 and Appendix §D.

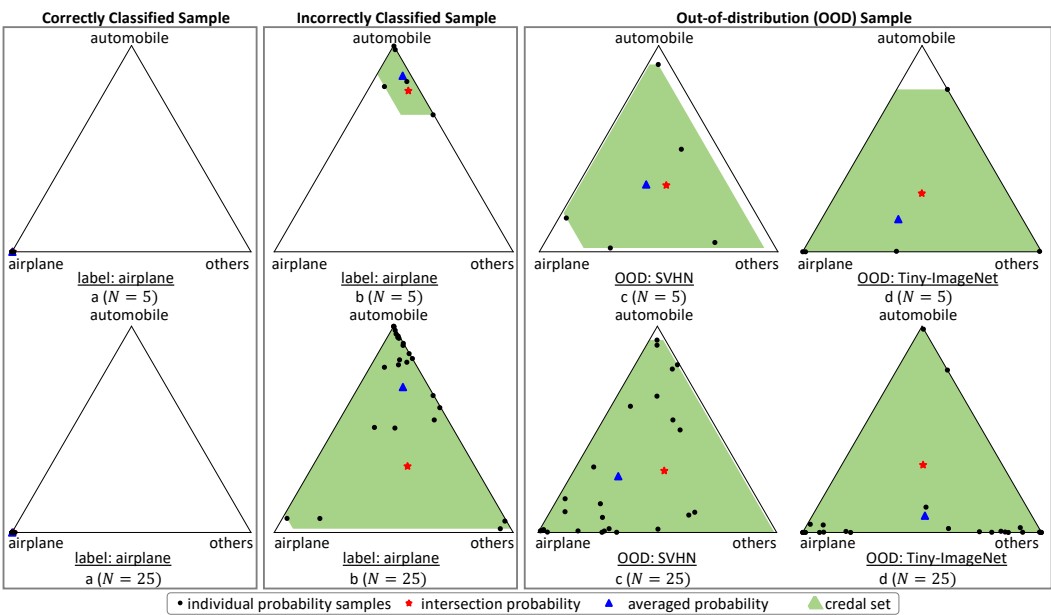

Figure 6: Visualization of our credal wrapper of ResNet50-based DEs on different ensemble sizes ($N = 5$ and $N = 25$) in three cases: correctly classified in-distribution (ID) sample (left), incorrectly classified sample (middle), and OOD samples (right).

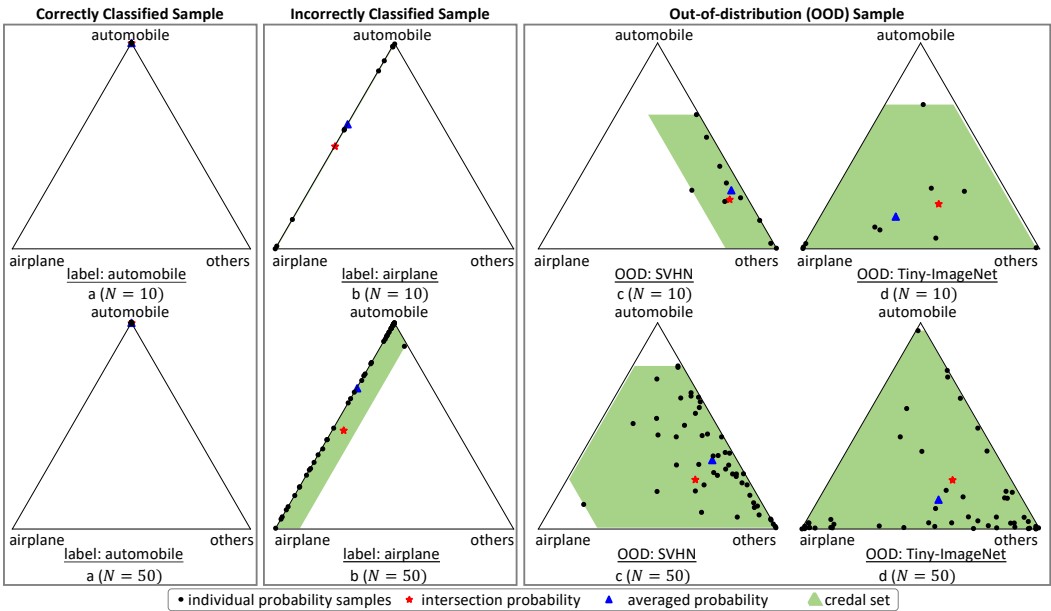

Figure 7: Visualization of our credal wrapper of VGG16-based BNNRs on different sample sizes ($N = 10$ and $N = 50$) in three cases: correctly classified in-distribution (ID) sample (left), incorrectly classified sample (middle), and OOD samples (right).

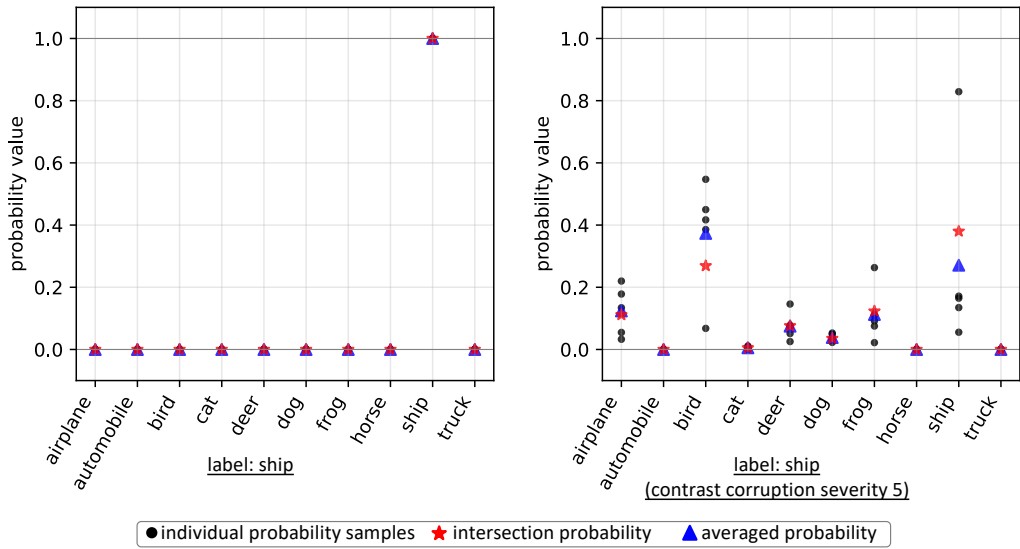

Figure 8: Visualization of the individual probabilities, the intersection, and the averaged probability for one CIFAR10 sample and its corrupted version sample. For the corrupted data, the intersection probability can remain the correct prediction while the averaged probability leads to a wrong prediction.

## A.2   ABLATION STUDY ON NUMBERS OF PREDICTIVE SAMPLES IN DES

In this experiment, we train 30 ResNet-50-based SNNs individually using the CIFAR10 dataset and under different random seeds. Then, we construct DEs using different numbers of ensemble members, namely $N = 3, 5, 10, 15, 20$ and $25$. Each type of DEs includes 15 instances using distinct seed combinations.

Table 5 reports OOD detection performance comparisons under different numbers of samples. The findings suggest that (i) Increasing the number of samples can indeed improve the EU estimation of our proposed credal wrapper. (ii) Compared to classical DEs, our method consistently shows superior performance on EU quantification, robustly against the number of samples. Figure 9 shows the ECE performance evaluation using different settings of $N$ on the CIFAR10-C samples. We can observe that: (i) Increasing the number of samples overall can lead to a lower ECE value; (ii) Compared to the naive averaging DE predictions, our intersection probability consistently achieves lower ECE values on corrupted instances.

Table 5: OOD detection performance (%) comparison in DEs.

| $N$ | CIFAR10 vs SVHN | | | | CIFAR10 vs Tiny-ImageNet | | | |
|---|---|---|---|---|---|---|---|---|
| | AUROC | | AUPRC | | AUROC | | AUPRC | |
| | Baseline | Ours | Baseline | Ours | Baseline | Ours | Baseline | Ours |
| 3 | 89.64±1.16 | **91.97±1.38** | 92.13±1.12 | **94.91±1.08** | 86.31±0.24 | **87.57±0.25** | 81.76±0.40 | **85.02±0.32** |
| 5 | 89.26±1.02 | **93.50±0.79** | 91.96±0.97 | **95.79±0.70** | 86.83±0.25 | **88.65±0.25** | 83.12±0.38 | **86.77±0.37** |
| 10 | 90.54±0.72 | **95.03±0.40** | 93.22±0.70 | **96.80±0.35** | 87.75±0.13 | **89.62±0.14** | 84.99±0.25 | **88.04±0.20** |
| 15 | 90.78±0.68 | **95.31±0.47** | 93.23±0.62 | **96.91±0.43** | 88.10±0.10 | **89.94±0.10** | 85.59±0.16 | **88.44±0.13** |
| 20 | 91.09±0.50 | **95.69±0.34** | 93.49±0.48 | **97.15±0.30** | 88.33±0.08 | **90.11±0.08** | 85.96±0.11 | **88.59±0.10** |
| 25 | 91.31±0.27 | **95.84±0.15** | 93.62±0.24 | **97.24±0.12** | 88.46±0.04 | **90.22±0.05** | 86.17±0.08 | **88.72±0.07** |

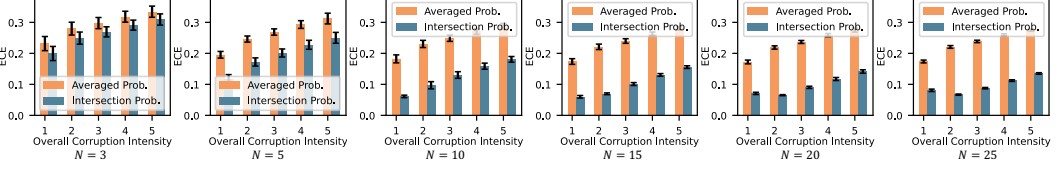

Figure 9: ECE values of DEs with various $N$ on CIFAR10-C against increased corruption intensity.

### A.3 Ablation Study on Numbers of Predictive Samples in BNNs

In this experiment, we first increase the sampling size of BNNs at prediction time, namely $N = 10$ and $N = 50$. Table 6 reports the OOD detection performance of BNNR and BNNF involving CIFAR10 (ID) vs SVHN and Tiny-ImageNet (OODs), based on the VGG16 backbone. It shows that the credal wrapper consistently produces better EU estimates, as evidenced by enhanced OOD detection performance. Further, Table 13 in the Appendix reports the same comparison based on the ResNet-18 architecture, confirming those results.

Table 6: OOD detection comparisons using EU (%) of VGG16-based BNNs.

| $N$ | Method | BNNR: CIFAR10 (ID) | | | | BNNF: CIFAR10 (ID) | | | |
|---|---|---|---|---|---|---|---|---|---|
| | | SVHN (OOD) | | Tiny-ImageNet (OOD) | | SVHN (OOD) | | Tiny-ImageNet (OOD) | |
| | | AUROC | AUPRC | AUROC | AUPRC | AUROC | AUPRC | AUROC | AUPRC |
| 10 | Baseline | 86.60±1.28 | 90.54±1.00 | 85.00±0.27 | 80.90±0.42 | 86.67±0.54 | 90.64±0.71 | 84.78±0.18 | 80.51±0.38 |
| | Ours | **88.38±1.53** | **92.61±1.12** | **86.28±0.25** | **83.88±0.32** | **88.03±0.52** | **92.55±0.52** | **85.79±0.18** | **83.29±0.28** |
| 50 | Baseline | 86.60±1.37 | 90.44±1.02 | 85.41±0.21 | 81.79±0.30 | 86.63±0.60 | 90.56±0.79 | 85.10±0.19 | 81.28±0.34 |
| | Ours | **88.90±1.65** | **92.75±1.22** | **87.17±0.26** | **85.22±0.33** | **88.60±0.56** | **92.91±0.54** | **86.61±0.21** | **84.64±0.27** |

Tables 5 and 6 suggest that our credal wrapper provides better uncertainty estimation than Bayesian methods, even when the latter uses more samples. For instance, the OOD detection performance of traditional DEs with $N = 25$ is significantly lower than that with $N = 5$ when using our credal wrapper. Moreover, the OOD performance of our credal wrapper with $N = 10$ consistently surpasses that of conventional BNNs with $N = 50$. From this perspective, our credal wrapper can reduce the complexity as increasing the number of samples typically demands more computational resources and memory, such as the additional single models required for DEs during training and inference.

Table 7: Performance comparison when applying $N = 1000$ to the ResNet-18-based BNNR.

| | ID Performance | | CIFAR10 vs SVHN | | CIFAR10 vs Tiny-ImageNet | |
|---|---|---|---|---|---|---|
| | ACC | ECE | AUROC | AUPRC | AUROC | AUPRC |
| Baseline | 91.89±0.18 | 0.057±0.002 | 87.24±1.72 | 93.09±0.89 | 83.94±0.32 | 81.15±0.39 |
| Ours | **91.90±0.16** | **0.034±0.003** | **89.36±1.25** | **93.88±0.86** | **86.89±0.19** | **84.63±0.33** |

Additionally, we further increase the number of samples ($N = 1000$) for the ResNet18-based BNNR to explore an extreme case. Table 7 compares the test ACC and ECE values between two approaches, namely $\tilde{p}$ and $p^*$, as well as the OOD detection performance. Although a high sampling size is rarely employed in BNNs due to the inherent complexity, the results illustrate the stability of our methods and the consistent superiority of our credal wrapper compared to the BNN baseline.

### A.4 Generalized Hartley Measure for EU Estimation of Credal Wrapper

In this section, we evaluate the EU quantification performance of our credal wrapper using the generalized Hartley (GH) measure (Abellán & Moral, 2000), and briefly discuss the interpretation of the generalized entropy and the GH measure in uncertainty quantification of the credal set.

**Definition and Implementation** The generalized Hartley measure (Abellán & Moral, 2000; Hüllermeier & Waegeman, 2021), $GH(\mathbb{P})$, captures the non-specificity across the distributions in the credal set, and can be seen as a proxy for its volume (Hüllermeier & Waegeman, 2021; Sale et al., 2023a) Mathematically, $GH(\mathbb{P})$ calculates the expectation of the Hartley measure over all possible subsets $\mathbb{B}$ on the target space $\mathbb{Y}$, as follows (Abellán & Moral, 2000):

$$GH(\mathbb{P}) = \sum_{\mathbb{B} \subseteq \mathbb{Y}} m_{\mathbb{P}}(\mathbb{B}) \cdot \log_2(|\mathbb{B}|), \tag{12}$$

in which $m_{\mathbb{P}}$ denotes the mass assignment function associated to a credal set $\mathbb{P}$ and $|\mathbb{B}|$ indicates the cardinality of $\mathbb{B}$. $m_{\mathbb{P}}(\mathbb{B})$ can be computed using the Möbius inverse of the capacity function $\nu_{\mathbb{P}}$ (Chateauneuf & Jaffray, 1989) as follows:

$$m_{\mathbb{P}}(\mathbb{B}) = \sum_{\mathbb{A} \subseteq \mathbb{B}} (-1)^{|\mathbb{B} \setminus \mathbb{A}|} \nu_{\mathbb{P}}(\mathbb{A}), \tag{13}$$

where $\mathbb{B}\setminus\mathbb{A} = y|y \in \mathbb{B}$ and $y \notin \mathbb{A}$ and $\nu_{\mathbb{P}}$ describes the lower probability of all possible subsets $\mathbb{A} \subseteq \mathbb{B}$. GH Measure is computationally complex to compute. However, in our case, the lower probability $\nu_{\mathbb{P}}(\mathbb{A})$ associated with the predicted credal set can be readily computed as follows:

$$\nu_{\mathbb{P}}(\mathbb{A}) = \max\left( \sum_{y_j \in \mathbb{A}} p_{L_j}, 1 - \sum_{y_j \notin \mathbb{A}} p_{U_j} \right), \tag{14}$$

where $p_L$ and $p_U$ are the lower and upper probability values per class in the defined credal set.

**Experimental Validation** In this ablation study, we evaluate on EU estimation quality of our credal wrapper using GH($\mathbb{P}$) measure. Table 8 reports OOD detection performance tested on CIFAR10 (ID) vs SVHN (OOD) and Tiny-ImageNet (OOD). All models are implemented on the VGG16 backbones. The results demonstrate that our credal wrapper consistently enhances EU estimation performance and is agnostic in the sense that it can accommodate any EU measure for credal sets.

Table 8: OOD detection AUROC and AUPRC performance (%) comparison between classical and credal wrapper of BNNs and DEs using EU. The results are from 15 runs.

| | Model | | EU | SVHN (OOD) | | Tiny-ImageNet (OOD) | |
|---|---|---|---|---|---|---|---|
| | | | | AUROC | AUPRC | AUROC | AUPRC |
| VGG16 | BNNR | Baseline | $H(\bar{p}) - \tilde{H}(p)$ | 86.65±1.26 | 90.61±0.88 | 84.62±0.28 | 80.06±0.40 |
| | | Ours | GH($\mathbb{P}$) | **87.24±1.32** | **91.54±0.97** | **84.88±0.28** | **81.05±0.37** |
| | BNNF | Baseline | $H(\bar{p}) - \tilde{H}(p)$ | 86.79±0.47 | 90.76±0.55 | 84.54±0.20 | 79.91±0.35 |
| | | Ours | GH($\mathbb{P}$) | **87.22±0.45** | **91.49±0.49** | **84.72±0.20** | **80.72±0.32** |
| | DEs | Baseline | $H(\bar{p}) - \tilde{H}(p)$ | 89.74±1.31 | 93.58±0.97 | 88.49±0.17 | 85.79±0.35 |
| | | Ours | GH($\mathbb{P}$) | **90.37±1.33** | **94.21±0.89** | **88.98±0.13** | **86.88±0.26** |

**Further Discussion on Different Uncertainty Measures** Given a credal set $\mathbb{P}$, it was suggested to assume the following representation to express the uncertainty (Abellán & Moral, 2000; Hüllermeier & Waegeman, 2021):

$$TU(\mathbb{P}) = EU(\mathbb{P}) + AU(\mathbb{P}). \tag{15}$$

As a result, given $TU(\mathbb{P})$ and a generalized measure of aleatoric uncertainty (conflict) $AU(\mathbb{P})$, a generalized measure of epistemic uncertainty (non-specificity) $EU(\mathbb{P})$ can be derived via disaggregation as $EU(\mathbb{P}) = TU(\mathbb{P}) - AU(\mathbb{P})$. The lower entropy $\underline{H}(\mathbb{P})$ applied in our work indicates the lower bound of the (aleatoric) uncertainty that remains even when all epistemic uncertainty is removed, i.e., when $\mathbb{P}$ is reduced to a single distribution $p \in \mathbb{P}$. From this perspective, $\underline{H}(\mathbb{P})$ corresponds in fact to the natural measure of irreducible (aleatoric) uncertainty. Study (Abellán et al., 2006) justifies the use of upper entropy as a global uncertainty measure for credal sets. As for the epistemic uncertainty, the difference $\overline{H}(\mathbb{P}) - \underline{H}(\mathbb{P})$ appears to be reasonable and refers to the size of the set of candidate entropies $[\underline{H}(\mathbb{P}), \overline{H}(\mathbb{P})]$, quantifying uncertainty about the aleatoric uncertainty $H(p)$ of $p$. Different from $\overline{H}(\mathbb{P}) - \underline{H}(\mathbb{P})$ in EU estimation, the GH($\mathbb{P}$) measures the size of the set of candidate probabilities, and hence refers to uncertainty or imprecision about the ground-truth probability. For further discussions on the different measures and their corresponding strengths and weaknesses in measuring predictive uncertainty, we kindly refer to (Hüllermeier et al., 2022).

## A.5 TOTAL UNCERTAINTY ESTIMATION EVALUATION ON OOD DETECTION BENCHMARKS

In this section, we assess the total uncertainty (TU) estimation quality (as opposed to EU) of our credal wrapper on various OOD detection benchmarks. Tables 9, 10, 11, 12, 13, and Figures 10, 11 show that our credal wrapper can improve the quality of TU estimation as well, across different dataset pairs, network architectures, and test settings.

## A.6 ABLATION STUDY ON OVERCONFIDENCE REGIME

In this section, we examine the performance of our credal wrapper in scenarios where the ensemble member (SNN) generates overconfident predictions. In our credal wrapper framework, overconfidence in one of the predictions would stretch the resulting credal set. This is because the upper and lower probability bounds over classes are derived via a max/min operation in eq. (5). For instance, in the case of 3 classes, if there are three distinct extreme probability vectors (three vertices of the simplex), our credal wrapper method will effectively convey complete uncertainty, with the resulting

Table 9: OOD detection AUROC and AUPRC performance (%) comparison between the classical and credal wrapper version of BNNs and DEs, using TU as the uncertainty metric. All models are implemented on VGG16/ResNet-18 backbones and tested on CIFAR10 (ID) vs SVHN (OOD) and Tiny-ImageNet (OOD). The results are from 15 runs. The best scores are in bold.

| | | ResNet-18: CIFAR10 (OOD) | | | | VGG16: CIFAR10 (OOD) | | | |
| | | SVHN (OOD) | | Tiny-ImageNet (OOD) | | SVHN (OOD) | | Tiny-ImageNet (OOD) | |
| | | AUROC | AUPRC | AUROC | AUPRC | AUROC | AUPRC | AUROC | AUPRC |
|---|---|---|---|---|---|---|---|---|---|
| BNNR | Baseline | 88.18±1.30 | 92.68±0.86 | 86.40±0.24 | 83.16±0.40 | 88.57±1.47 | 93.26±1.08 | 85.50±0.31 | 82.60±0.41 |
| | Ours | **88.36±1.30** | **92.95±0.87** | **86.52±0.22** | **83.66±0.38** | **88.79±1.57** | **93.44±1.14** | **85.89±0.31** | **83.53±0.43** |
| BNNF | Baseline | 88.43±0.83 | 92.90±0.51 | 86.45±0.33 | 83.22±0.33 | 88.32±0.50 | 93.06±0.51 | 85.30±0.20 | 82.32±0.30 |
| | Ours | **88.65±0.81** | **93.19±0.51** | **86.58±0.32** | **83.75±0.31** | **88.47±0.52** | **93.27±0.52** | **85.62±0.20** | **83.23±0.28** |
| DE | Baseline | 88.47±0.80 | 92.82±0.64 | 88.96±0.14 | 87.22±0.18 | 91.20±1.29 | 94.87±0.78 | 89.55±0.08 | 87.89±0.14 |
| | Ours | **88.98±0.86** | **93.13±0.70** | **89.53±0.14** | **88.23±0.17** | **92.12±1.32** | **95.41±0.75** | **90.00±0.07** | **88.49±0.17** |

Table 10: OOD detection AUROC and AUPRC performance (%) of both the classical and credal wrapper version of DEs using TU as the metric. The results are from 15 runs, based on the ResNet-50 backbone. The best scores are in bold.

| | CIFAR10 (ID) | | | | CIFAR100 (ID) | | | | ImageNet (ID) | |
| | SVHN (OOD) | | Tiny-ImageNet (OOD) | | SVHN (OOD) | | Tiny-ImageNet (OOD) | | ImageNet-O (OOD) | |
| | AUROC | AUPRC | AUROC | AUPRC | AUROC | AUPRC | AUROC | AUPRC | AUROC | AUPRC |
|---|---|---|---|---|---|---|---|---|---|---|
| Baseline | 94.80±0.43 | 97.26±0.29 | 88.80±0.19 | 87.21±0.29 | 78.53±1.94 | 88.83±1.01 | 80.75±0.15 | 77.65±0.19 | 50.20±0.07 | 50.44±0.06 |
| Ours | **95.44±0.37** | **97.57±0.23** | **89.30±0.17** | **87.97±0.25** | **80.71±1.96** | **89.97±0.99** | **81.46±0.14** | **78.29±0.17** | **54.87±0.08** | **52.27±0.05** |

Table 11: OOD detection AUROC and AUPRC performance (%) of both the classical and credal wrapper version of DEs using TU as the metric. Results are from 15 runs, based on EffB2 and ViT-B backbones. The best scores are in bold.

| | | CIFAR10 (ID) | | | | CIFAR100 (ID) | | | |
| | | SVHN (OOD) | | Tiny-ImageNet (OOD) | | SVHN (OOD) | | Tiny-ImageNet (OOD) | |
| | | AUROC | AUPRC | AUROC | AUPRC | AUROC | AUPRC | AUROC | AUPRC |
|---|---|---|---|---|---|---|---|---|---|
| EffB2 | Baseline | 97.55±0.27 | 98.78±0.16 | 93.41±0.15 | 93.01±0.17 | 88.46±1.33 | 94.40±0.59 | 86.45±0.10 | **84.88±0.14** |
| | Ours | **97.91±0.21** | **98.98±0.10** | **93.74±0.15** | **93.55±0.16** | **89.35±1.20** | **94.83±0.52** | **86.61±0.09** | 84.87±0.12 |
| ViT-B | Baseline | 79.80±1.75 | 87.97±1.17 | 83.81±0.81 | 81.67±0.89 | 84.10±1.12 | 91.41±0.72 | 82.64±0.28 | 79.94±0.43 |
| | Ours | **81.08±1.82** | **88.48±1.48** | **84.62±0.75** | **82.82±0.85** | **85.40±1.05** | **92.02±0.70** | **83.06±0.29** | **80.34±0.41** |

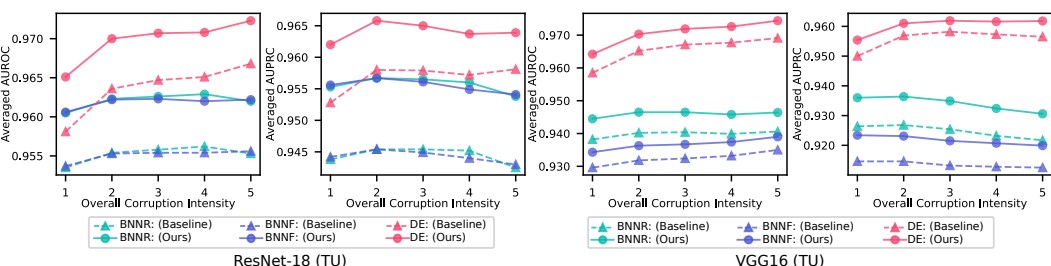

Figure 10: OOD detection using TU as metric on CIFAR10 vs CIFAR10-C of both the classical and credal wrapper version BNNs and DE against increased corruption intensity, using VGG16 and ResNet-18 as backbones.

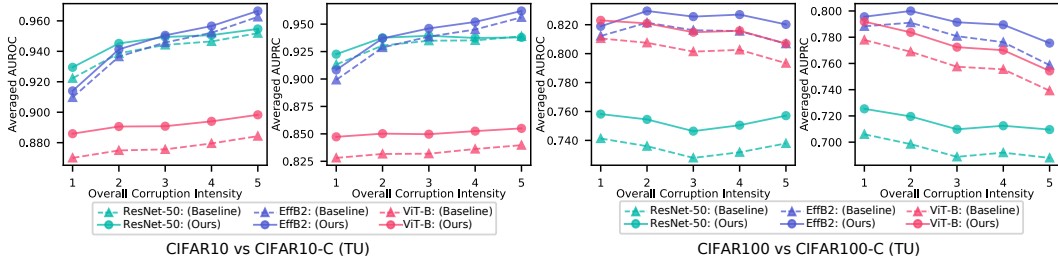

Figure 11: OOD detection using TU as the metric on CIFAR10/100 vs CIFAR10-C/100-C of both the classical and credal wrapper version of DEs against increased corruption intensity, using ResNet-50, EffB2, and ViT-B as backbones.

Table 12: Ablation study on numbers of predictive samples in DEs: OOD detection AUROC and AUPRC performance (%) of both the classical and credal wrapper version of DEs using TU as uncertainty metrics, involving CIFAR10 (ID) vs SVHN (OOD) and Tiny-ImageNet (OOD). The results are from 15 runs. The best scores are in bold.

| | CIFAR10 vs SVHN | | | | CIFAR10 vs Tiny-ImageNet | | | |
|---|---|---|---|---|---|---|---|---|
| | AUROC | | AUPRC | | AUROC | | AUPRC | |
| $N$ | Baseline | Ours | Baseline | Ours | Baseline | Ours | Baseline | Ours |
| 3 | 94.30±0.99 | **94.68±1.01** | 97.03±0.62 | **97.21±0.65** | 88.24±0.23 | **88.54±0.23** | 86.46±0.33 | **86.98±0.32** |
| 5 | 94.78±0.54 | **95.40±0.55** | 97.19±0.38 | **97.49±0.40** | 88.71±0.25 | **89.17±0.26** | 87.10±0.37 | **87.86±0.37** |
| 10 | 95.47±0.34 | **96.18±0.31** | 97.61±0.23 | **97.91±0.23** | 89.27±0.13 | **89.86±0.14** | 87.90±0.19 | **88.63±0.19** |
| 15 | 95.55±0.32 | **96.25±0.37** | 97.63±0.23 | **97.88±0.29** | 89.48±0.08 | **90.11±0.10** | 88.19±0.12 | **88.87±0.12** |
| 20 | 95.72±0.23 | **96.46±0.26** | 97.74±0.15 | **97.98±0.23** | 89.60±0.07 | **90.24±0.08** | 88.34±0.09 | **88.95±0.11** |
| 25 | 95.80±0.12 | **96.53±0.12** | 97.78±0.08 | **98.01±0.10** | 89.67±0.04 | **90.32±0.05** | 88.42±0.05 | **89.01±0.07** |

Table 13: Ablation study on numbers of predictive samples in BNNs: OOD detection AUROC and AUPRC performance (%) of both the classical and credal wrapper version of BNNs with increased number of samples, involving CIFAR10 (ID) vs SVHN (OOD) and Tiny-ImageNet (OOD). The results are from 15 runs and the best scores per uncertainty metric are in bold.

| Model | $N$ | Method | EU Measure as Metric | | | | TU Measure as Metric | | | |
|---|---|---|---|---|---|---|---|---|---|---|
| | | | SVHN (OOD) | | Tiny-ImageNet (OOD) | | SVHN (OOD) | | Tiny-ImageNet (OOD) | |
| | | | AUROC | AUPRC | AUROC | AUPRC | AUROC | AUPRC | AUROC | AUPRC |
| VGG16 | 10 | Baseline | 86.60±1.28 | 90.54±1.00 | 85.00±0.27 | 80.90±0.42 | 88.63±1.49 | 93.26±1.10 | 85.77±0.27 | 82.93±0.35 |
| | | Ours | **88.38±1.53** | **92.61±1.12** | **86.28±0.25** | **83.88±0.32** | **88.99±1.61** | **93.45±1.17** | **86.44±0.27** | **84.28±0.34** |
| BNNR | 50 | Baseline | 86.60±1.37 | 90.44±1.02 | 85.41±0.21 | 81.79±0.30 | 88.67±1.54 | **93.29±1.12** | 86.02±0.24 | 83.31±0.30 |
| | | Ours | **88.90±1.65** | **92.75±1.22** | **87.17±0.26** | **85.22±0.33** | **89.17±1.69** | 93.24±1.25 | **87.22±0.26** | **85.35±0.33** |
| BNNF | 10 | Baseline | 86.67±0.54 | 90.64±0.71 | 84.78±0.18 | 80.51±0.38 | 88.31±0.50 | 93.05±0.50 | 85.44±0.19 | 82.53±0.34 |
| | | Ours | **88.03±0.52** | **92.55±0.52** | **85.79±0.18** | **83.29±0.28** | **88.58±0.53** | **93.31±0.51** | **85.99±0.19** | **83.80±0.29** |
| | 50 | Baseline | 86.63±0.60 | 90.56±0.79 | 85.10±0.19 | 81.28±0.34 | 88.34±0.53 | 93.07±0.51 | 85.63±0.20 | 82.80±0.32 |
| | | Ours | **88.60±0.56** | **92.91±0.54** | **86.61±0.21** | **84.64±0.27** | **88.86±0.55** | **93.34±0.54** | **86.67±0.21** | **84.82±0.27** |
| ResNet-18 | 10 | Baseline | 88.46±1.23 | 93.19±0.77 | 85.85±0.22 | 81.58±0.37 | 88.24±1.29 | 92.71±0.85 | 86.43±0.24 | 83.22±0.40 |
| | | Ours | **88.61±1.26** | **93.35±0.83** | **86.16±0.20** | **82.99±0.26** | **88.53±1.29** | **93.09±0.88** | **86.63±0.21** | **83.91±0.35** |
| BNNR | 50 | Baseline | 88.43±1.27 | 93.33±0.77 | 85.58±0.23 | 81.70±0.42 | 88.28±1.27 | 92.74±0.84 | 86.45±0.24 | 83.25±0.40 |
| | | Ours | **88.93±1.24** | **93.61±0.83** | **86.48±0.21** | **83.80±0.34** | **88.84±1.26** | **93.35±0.86** | **86.82±0.22** | **84.38±0.36** |
| BNNF | 10 | Baseline | 88.77±0.85 | 93.45±0.56 | 85.97±0.31 | 81.83±0.27 | 88.49±0.83 | 92.93±0.50 | 86.48±0.34 | 83.26±0.35 |
| | | Ours | **88.96±0.83** | **93.62±0.55** | **86.28±0.33** | **83.22±0.36** | **88.84±0.81** | **93.36±0.52** | **86.70±0.33** | **84.03±0.35** |
| | 50 | Baseline | 88.79±0.95 | 93.59±0.61 | 85.67±0.35 | 81.88±0.35 | 88.54±0.82 | 92.97±0.50 | 86.49±0.33 | 83.30±0.35 |
| | | Ours | **89.30±0.83** | **93.89±0.55** | **86.59±0.33** | **83.99±0.35** | **89.18±0.81** | **93.64±0.54** | **86.90±0.32** | **84.53±0.36** |

credal set encompassing the entire simplex. This conservative nature can be sensible, as it expresses our full ignorance of the correct classification.

Furthermore, we conduct an additional ablation study to explore the benefit of our method given overconfident prediction samples. We manually and randomly transform one of the predictions of the same input image from five deep ensembles to a one-hot probability and compare the OOD detection performance of our method and the classical approach. Table 14 also shows our credal wrapper can improve the uncertainty quantification even in overconfident scenarios.

Table 14: OOD detection using EU (left) and TU (right) as uncertainty metrics in overconfident scenarios. The results are from 15 runs, based on the ResNet-50 backbone. The best scores per uncertainty metric are in bold.

| Model | Method | Epistemic Uncertainty Measure as Metric | | | | Total Uncertainty Measure as Metric | | | |
| | | SVHN (OOD) | | Tiny-ImageNet (OOD) | | SVHN (OOD) | | Tiny-ImageNet (OOD) | |
| | | AUROC | AUPRC | AUROC | AUPRC | AUROC | AUPRC | AUROC | AUPRC |
| DE | Baseline | 90.42±0.81 | 93.30±0.86 | 87.01±0.20 | 83.53±0.28 | 94.36±0.45 | 96.98±0.32 | 88.54±0.19 | 86.82±0.30 |
| | Ours | **94.92±0.39** | **97.26±0.25** | **88.98±0.17** | **87.42±0.27** | **95.24±0.38** | **97.48±0.23** | **89.12±0.17** | **87.74±0.27** |

## A.7 QUALITATIVE EVALUATION OF UNCERTAINTY ESTIMATION OF CREDAL WRAPPER

In this section, we present the estimates of the EU and TU on the ID and OOD samples of the classical DE (using $H(\tilde{p}) - \tilde{H}(p)$ and $H(\tilde{p})$, respectively) and our credal wrapper (using $\overline{H}(\mathbb{P}) - \underline{H}(\mathbb{P})$ and $\overline{H}(\mathbb{P})$, respectively) in Figures 12 and 13. The results qualitatively demonstrate that our credal wrapper consistently enhances the uncertainty quantification of classical DEs. This is evidenced by the significantly larger uncertainty estimates for the OOD samples, which correspond to the expected and desirable behavior.

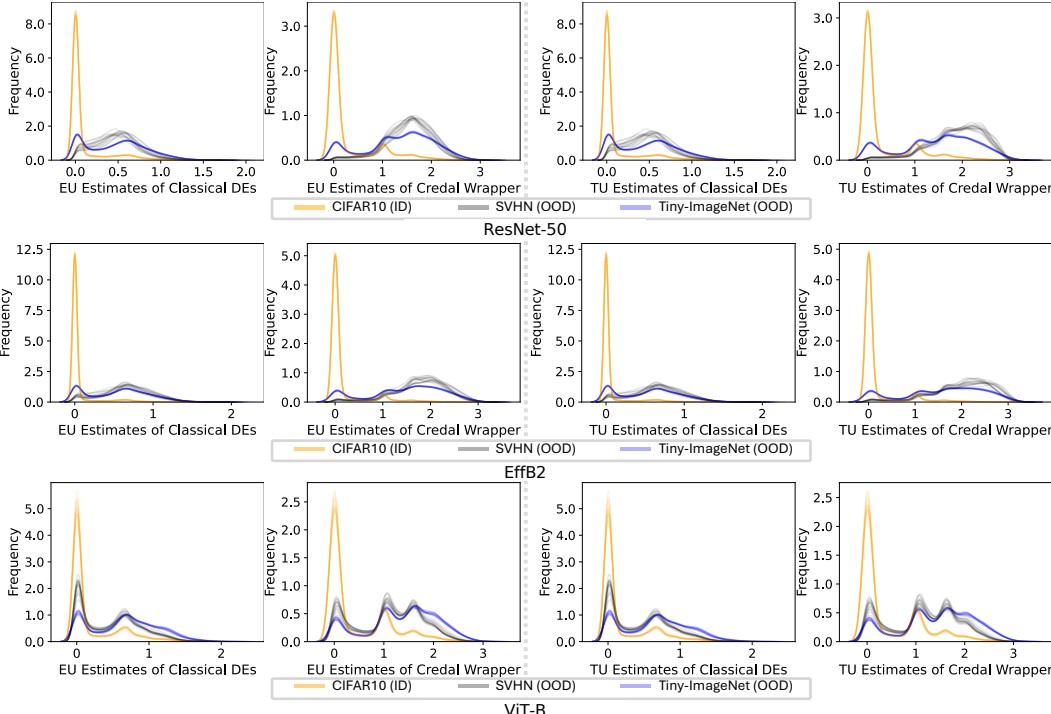

Figure 12: EU and TU estimates of ID (CIFAR10) and OOD (SVHN and Tiny-ImageNet) samples of the classical and credal wrapper version of DEs, obtained using ResNet-50, EffB2, and ViT backbones. Results are from 15 runs.

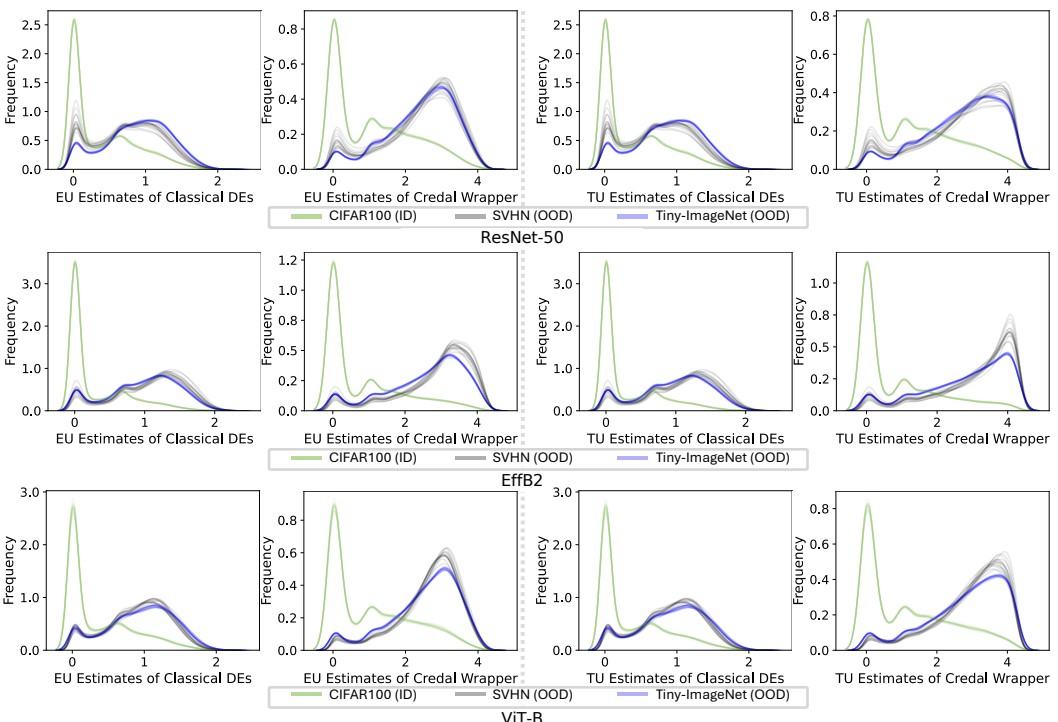

Figure 13: EU and TU estimates of ID (CIFAR100) and OOD (SVHN and Tiny-ImageNet) samples of the classical and credal wrapper version of DEs, obtained using ResNet-50, EffB2, and ViT backbones. Results are from 15 runs.

## A.8 EVALUATION OF INTERSECTION PROBABILITY ON CORRUPTED SAMPLES USING NLL METRIC

In this section, we further evaluate the intersection probability on corrupted samples using the negative log-likelihood (NLL) metric. A smaller NLL indicates that the model is more confident and accurate in predicting the correct class for each input (Dusenberry et al., 2020). Figures 14 and 15 show the consistent superiority of the intersection probability on corrupted data in extensive test cases, as evidenced by smaller NLL values.

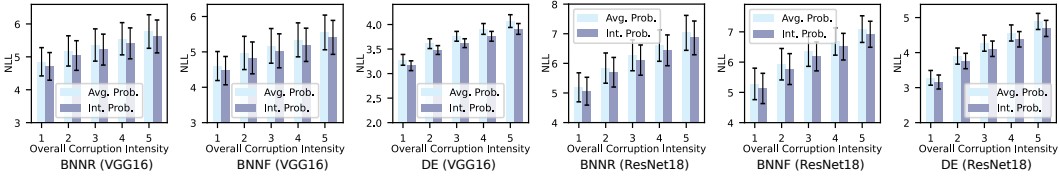

Figure 14: NLL values of BNNR, BNNF, and DE on CIFAR10-C against increased corruption intensity, using the averaged probability (Avg. Prob.) and our proposed intersection probability (Int. Prob.). VGG16 and ResNet-18 are backbones. Results are from 15 runs.

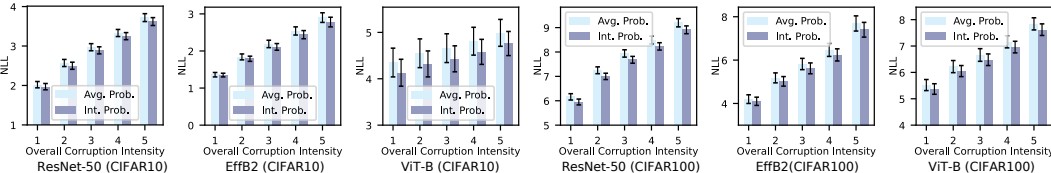

Figure 15: NLL values of DEs on CIFAR10-C and CIFAR100-C against increased corruption intensity, using the averaged probability (Avg. Prob.) and our proposed intersection probability (Int. Prob.). ResNet-50, EffB2, and ViT-B are backbones. Results are from 15 runs.

## A.9 Additional Results on Computational Cost

In this section, we present the CPU time cost comparison in Table 15. The results demonstrate that, despite the higher time cost, our method remains practical, potentially even achieving real-time capability, particularly for low-dimensional classification problems without strict computational resource constraints. In addition, we are confident that a more efficient code implementation of our approach could further reduce computational overhead.

Table 15: Time cost (ms) comparison for uncertainty (entropy) calculation of a single input instance. The cost is measured by a single Intel(R) Xeon(R) Gold 6240 CPU @ 2.60GHz and is averaged from 20 runs.

| $C$ | 3 | 10 | 20 | 100 |
|---|---|---|---|---|
| Baseline | 0.002 | 0.002 | 0.002 | 0.006 |
| Ours | 2.526 | 5.921 | 9.694 | 87.384 |

Furthermore, our method demonstrates superior uncertainty quantification performance while requiring fewer predictive samples ($N$), as shown in Tables 5 and 6. For instance, the credal set version of DEs-5 ($N = 5$) surpasses the classical DEs-25 in performance. Similarly, the credal set versions of BNNR and BNNs achieve higher performance using only 10 predictive samples compared to their classical counterparts, which require 50 samples. We report the inference complexity of these models across various values of $N$ in Tables 16 and 17. The inference times are measured using the same single CPU and an advanced Nvidia A100-SXM4-80GB GPU. The results indicate that the additional overhead introduced by our method for entropy calculation is negligible, especially when considering the substantial reduction in overall inference cost. In addition, our approach significantly reduces GPU memory consumption. For example, DEs-25 and BNNF ($N = 50$) fail to make inferences on a single Nvidia A100-SXM4-40GB GPU due to memory constraints.

From the perspective of delivering higher uncertainty quantification performance with fewer predictive samples, the proposed credal wrapper reduces the computational complexity and enhances practical applicability.

Table 16: Inference complexity for a single input instance from ResNet-50-based DEs with different ensemble sizes. The inference time is averaged from 20 runs. CPU: A single Intel(R) Xeon(R) Gold 6240 CPU @ 2.60GHz; GPU: A single Nvidia A100-SXM4-80GB GPU.

| | Ensemble Size $N$ | Model Size (MB) | Inference CPU Cost (ms) | Inference GPU Cost (ms) |
|---|---|---|---|---|
| DEs | 5 | 500.05 | 941.9 ± 144.4 | 300.4 ± 97.7 |
| DEs | 25 | 2500.25 | 4495.3 ± 120.8 | 1429.7 ± 110.0 |

Table 17: Inference complexity for a single input instance from VGG16-based BNNRs and BNNFs with sample ensemble size. The inference time is averaged from 20 runs. CPU: A single Intel(R) Xeon(R) Gold 6240 CPU @ 2.60GHz; GPU: A single Nvidia A100-SXM4-80GB GPU.

| | Sample Size $N$ | Inference CPU Cost (ms) | Inference GPU Cost (ms) |
|---|---|---|---|
| BNNR | 10 | 4951.4 ± 91.8 | 1982.9 ± 204.3 |
| BNNR | 50 | 23466.4 ± 165.6 | 9758.2 ± 197.9 |
| BNNF | 10 | 5474.9 ± 92.7 | 2241.1 ± 203.8 |
| BNNF | 50 | 27498.8 ± 478.1 | 11046.2 ± 214.6 |

## A.10 Additional Results on Different Hyperparameters of PIA Algorithm

The hyperparameter $J$ of the PIA algorithm 1 is a design choice intended to mitigate the computational cost associated with uncertainty quantification, particularly in situations with limited computational resources. As demonstrated in the ablation study results in Table 4, the proposed PIA algorithm significantly reduces the computational complexity of uncertainty estimation in high-class classification scenarios, while maintaining strong uncertainty performance on OOD detection

benchmarks. Increasing the value of $J$ enhances the performance. Furthermore, we present the EU and TU estimates for ID (CIFAR100) and OOD (SVHN and Tiny-ImageNet) samples at different values of $J$ in Figure 4. The figure shows that using a small value of $J$ leads to underestimating the EU and TU, particularly for OOD samples. This phenomenon explains why a higher value of $J$ can achieve better OOD detection performance.

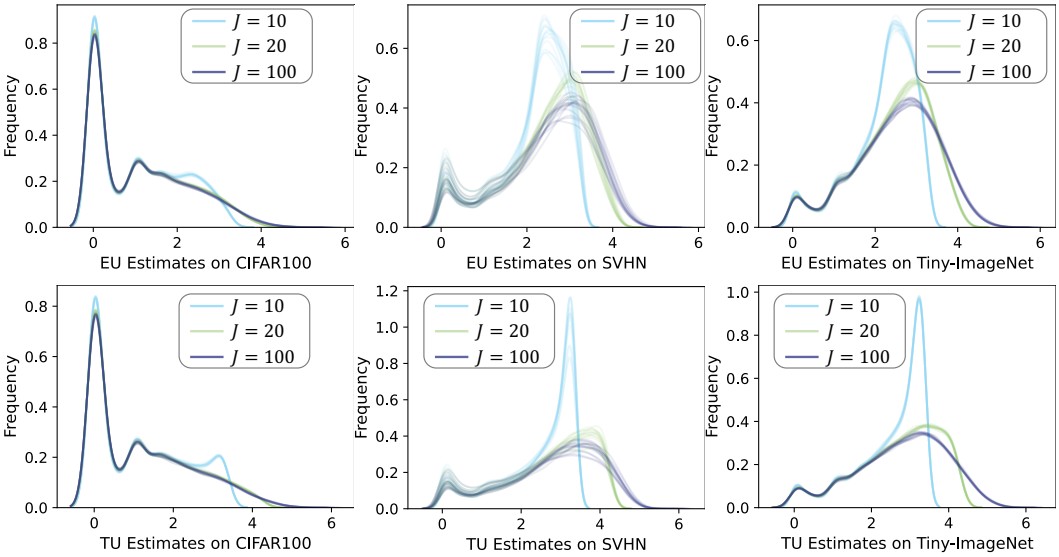

Figure 16: EU and TU estimates of ID (CIFAR100) and OOD (SVHN and Tiny-ImageNet) samples of the credal wrapper version of DEs using different values of $J$, obtained using the ResNet-50 backbone. Results are from 15 runs.

Table 18: OOD detection AUROC and AUPRC performance (%) of credal wrapper of DEs using TU as uncertainty metrics using different setting of $J$ of PIA algorithm. The OOD detection involves CIFAR100 (ID) vs SVHN (OOD) and Tiny-ImageNet (OOD). The results are from 15 runs, based on the ResNet-50 backbone.

| $J$ | TU as Metric | | | |
| | SVHN | | Tiny-ImageNet | |
| | AUROC | AUPRC | AUROC | AUPRC |
| 10 | 80.05±1.92 | 89.30±0.97 | 81.24±0.14 | 77.60±0.18 |
| 20 | 80.71±1.96 | 89.97±0.99 | 81.46±0.14 | 78.29±0.17 |
| 100 | 80.90±1.98 | 90.07±1.00 | 81.52±0.14 | 78.50±0.15 |

## B EXPERIMENTAL IMPLEMENTATION DETAILS

**Training Details** In terms of the evaluation using the small-scale datasets, all models are implemented on the established VGG16 and ResNet-18 architectures using the CIFAR10 dataset. The Adam optimizer is applied with a learning rate scheduler, initialized at 0.001, and subjected to a 0.1 reduction at epochs 80 and 120. Standard neural networks (SNNs) and BNNs are trained for 100 and 150 epochs, respectively. Standard data augmentation is uniformly implemented across all methodologies to enhance the training performance quality of training data and training performance. The training batch size is set as 128. The device is a single Tesla P100-SXM2-16GB GPU. The standard data split is applied. In terms of EDDs, for each learning rate scheduler configuration, as discussed in Sec. 4, we train 15 models for 100 epochs, one from each distinct DE. All other training configurations, such as data splitting and augmentation, are kept consistent with those used for the other models. Additionally, temperature annealing, outlined in the original EDD paper (Malinin et al., 2019), is also applied during the EDD training process.

In terms of the evaluation using the large-scale datasets, we mainly use two Tesla P100-SXM2-16GB GPUs as devices to contract deep ensembles by independently training 15 standard neural

networks (SNNs) under different random seeds. The input shape of the networks is (224, 224, 3). The Adam optimizer is employed, with a learning rate scheduler set at 0.001 and reduced to 0.0001 during the final 5 training epochs. Concerning the CIFAR10 dataset, SNNs are trained using 15, 15, and 25 epochs for ResNet-50, EffB2, and ViT-B backbones, respectively. As for the CIFAR100 dataset, SNNs are trained using 20, 20, and 25 epochs for ResNet-50, EffB2, and ViT-B backbones, respectively. Note that models based on ViT-B backbones are trained on one single Nvidia A100-SXM4-40GB GPU. In the ImageNet experiments, we employ one single Nvidia A100-SXM4-40GB GPU to retrain SNNs based on a pre-trained ResNet-50 model for 3 epochs, using the Adam optimizer with an initialized learning rate of $1e^{-6}$, under different random seeds. Standard data split is applied to all training processes. In ImageNet experiments, we apply three epochs as we observed that increasing the number of epochs further leads to lower validation loss and accuracy. The choice of not training them from scratch is mainly due to computational constraints. Although this setting may have implications for the performance of ensembles, regardless of the absolute baseline performance of ensembles under different training settings, our method exhibits a consistent improvement.

Table 19: Poor ID prediction and OOD detection performance of EDD-Fair. CIFAR10 as ID data.

| Model | | ID Prediction Performance | | OOD Detection Performance using EU | | | |
| | | | | SVHN (OOD) | | Tiny-ImageNet (OOD) | |
| | | ACC (%) | ECE | AUROC | AUPRC | AUROC | AUPRC |
| VGG16 | EDD-fair | 56.57±12.56 | 0.249±0.047 | 56.60±12.35 | 71.10±7.51 | 55.73±4.75 | 51.32±3.19 |
| ResNet-18 | EDD-fair | 85.68±3.48 | 0.102±0.056 | 85.30±7.85 | 90.16±5.95 | 76.84±4.59 | 69.84±4.29 |

**OOD Detection Process** In this paper, the OOD detection process is treated as a binary classification. We label ID and OOD samples as 0 and 1, respectively. The model's uncertainty estimation (using the EU or TU) for each sample is the 'prediction' for the detection. In terms of performance indicators, the applied AUROC quantifies the rates of true and false positives. The AUPRC evaluates precision and recall trade-offs, providing valuable insights into the model's effectiveness across different confidence levels.

**ECE Evaluation Process** In the context of ECE, a 'well-calibrated' prediction is expected to have a confidence value of 80% and be correct in approximately 80% of the test cases. To calculate ECE, predictions are split into a predetermined number $Q$ of bins $B$ of equal confidence range. The ECE is then calculated by summing the absolute difference between the average accuracy and confidence within each bin (Mehrtens et al., 2023):

$$\text{ECE} := \sum_{g=1}^{G} \frac{|B_g|}{n} \left| \text{acc}(B_g) - \text{conf}(B_g) \right|, \tag{16}$$

where $|B_g|$ is the number of samples in the $g$-th bin and $n$ is the total number of samples.

## C   Discussion on Generating Credal Sets from Individual Probability Distributions

Given a finite set of probability predictions, a possible alternative method for generating a credal set, termed *convex hull* approach, also exists. This method involves directly constructing the convex hull from the finite set of predictions. In our *credal wrapper*, the probability intervals define the credal set by injecting upper and lower bounds per class in eq. (5) and eq. (6), whereas the *convex hull* method requires computing all vertices of the credal sets.

The theoretical underpinning for the convex hull method when reasoning with coherent lower probabilities (and, therefore, the corresponding credal sets) is that it allows us to comply with the coherence principle (Walley, 1991). In a Bayesian context, individual predictions (such as those of networks with specified weights) can be interpreted as subjective pieces of evidence about a fact (e.g., what is the true class of an input observation). Coherence ensures that one realizes the full implications of such partial assessments (Walley, 1991; Cuzzolin, 2008). Figure 17 conceptually shows the differences between two methods in a 2D simplex.

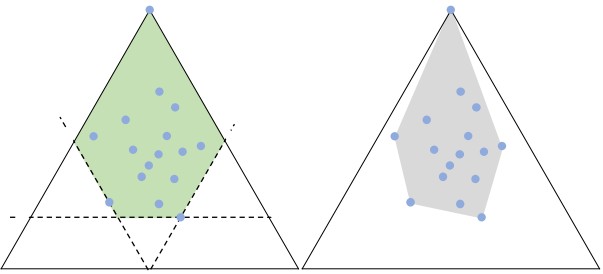

Figure 17: Different credal set generation methods. Left: our *credal wrapper*; right: *convex hull*.

Compared to the *convex hull* method, our probability interval systems exhibit a more conservative nature. Another practical difference is that the *convex hull* method is highly computationally complex, preventing it from being practically implemented in multi-class classification tasks. In the following, we aim to explain the associated complexity of the calculation process.

Given such a coherent lower probability, denoted as $\underline{P}$, the associated credal set $\mathbb{K}(\underline{P})$ comprises all probability measures ($\mathcal{P}(\mathbb{Y})$) in the target space ($\mathbb{Y}$) that can be defined as follows:

$$\mathbb{K}(\underline{P}) := \{P \in \mathcal{P}(\mathbb{Y}) \mid P(\mathbb{A}) \geq \underline{P}(\mathbb{A}) \; \forall \mathbb{A} \subseteq \mathbb{Y}\}. \tag{17}$$

Further, $\underline{P}$ is considered coherent if it can be computed as:

$$\underline{P}(\mathbb{A}) = \min_{P \in \mathbb{K}(\underline{P})} P(\mathbb{A}) \; \forall \mathbb{A} \subseteq \mathbb{Y} \tag{18}$$

where $\mathbb{A}$ denotes any possible subset of the target space $\mathbb{Y}$.

Mapping the finite set probabilistic predictions to a credal set generally requires the following steps:

i) Compute the lower probability for any subset $\mathbb{A}$ of the target space $\underline{P}(\mathbb{A})$.

ii) Compute the masses over the subsets by using Möbius inverse (Chateauneuf & Jaffray, 1989):

$$m_{\underline{P}}(\mathbb{A}) = \sum_{\mathbb{B} \subseteq \mathbb{A}} (-1)^{|\mathbb{A} \setminus \mathbb{B}|} \underline{P}(\mathbb{B}) \quad \forall \mathbb{A} \subseteq \mathbb{Y}. \tag{19}$$

iii) Compute the vertices of the credal set $\mathbb{K}(\underline{P})$. Its vertices are all the distributions $P^\pi$ induced by a permutation $\pi = \{x_{\pi(1)}, \ldots, x_{\pi(|\mathbb{Y}|)}\}$ of the elements of its sample space $\mathbb{Y}$, of the form (Chateauneuf & Jaffray, 1989; Cuzzolin, 2008)

$$P^\pi[\underline{P}](x_{\pi(i)}) = \sum_{\mathbb{A} \ni x_{\pi(i)}; \; \mathbb{A} \not\ni x_{\pi(j)} \; \forall j < i} m_{\underline{P}}(\mathbb{A}) \tag{20}$$

The vertex (the extreme probability) assigns to each class, put in position $\pi(i)$ by the permutation $\pi$, the mass of all the focal elements (sets of classes with non-zero mass) containing it, but not containing any elements preceding it in the permutation order (Wallner, 2005).

The above procedure is quite computationally expensive, because i) When the number of classes $|\mathbb{Y}| = C$ is large, it can be highly inefficient to compute probabilities for all $2^C$ events in the power set of $\mathbb{Y}$). ii) Compute the vertices in eq. (20) is extremely computationally difficult.

On the contrary, the probability interval systems applied do not suffer such computational burden, as shown in eq. (5) and eq. (6), the resulting credal set is directly defined by $C$ pairs probability bounds over classes. The extensive experimental validation also strongly demonstrates the consistently improved uncertainty performance of our proposed *credal wrapper*.

## D    DISCUSSION ON CREDAL WRAPPER IN CASES OF INFINITE SAMPLES

In this section, we aim to discuss the behaviour of the credal wrapper and the associated intersection probability in the case of infinitely many samples from individual probability distributions. In this

context, we assume that the support of the sampled distribution spans the entire simplex, without imposing any additional distributional constraints.

If we assume the predictions are sampled from a 'second-order' distribution (as in the BNN case), when one keeps collecting samples the corresponding credal set in eq. (6) expands (as it relies on min/max probability calculations in eq. (5)). Therefore, as $p^*$ is a function of the *probability interval bounds* of the credal set (and not the actually collected samples), for $N$ that goes to infinity will tend to the $p^*$ of the probability simplex $\Delta$, i.e., the uniform distribution. Indeed, this does not happen to the BMA theoretically, which averages the samples. However:

(i) The argument for the rationality of the representative $p^*$ of an interval probability system is compelling (Cuzzolin, 2009; 2022). The same asymptotic effect is shared by the center of mass of the credal set, indicating that this is a feature inherent to the credal representation itself.

(ii) In practice, $N$ is small even in the general cases of BNNs ($N = 5$). In the ablation study in Sec. A.3, we examine the performance of our credal wrapper in cases of $N = 10$, $N = 50$, and even $N = 1000$, our credal wrapper consistently shows an improved uncertainty estimation and calibration performance, as shown in Tables 7, 6 and Figures 3, 14. As the practical performance of $p^*$ is better than BMA, it appears that BMA can be biased/less representative at $N$ never going to infinite w.r.t. $p^*$, whereas it enjoys better convergence properties.

(iii) This does not apply to ensembles, as ensemble members are not 'sampled' from any distribution. The ablation study in Appendix §A.2 as well as Figures 3, 5, 14, 15 suggest that the consistent superior performance of $p^*$ of calibration performance on corrupted data.

