# OpenReview forum: "Credal Wrapper of Model Averaging for Uncertainty Estimation in Classification"
_ICLR.cc/2025/Conference — ICLR 2025 Spotlight_

### Official Review · Reviewer_Err5 · 2024-10-24

**Soundness:** 2
**Presentation:** 3
**Contribution:** 1
**Rating:** 5
**Confidence:** 3

**Summary:**

This paper proposes the Credal Wrapper, an ensemble aggregation method designed to improve uncertainty estimation. By applying the concept of credal set representation to neural network ensembles, the algorithm estimates probability intervals and intersection probabilities. Experimental results demonstrate that the proposed method enhances the performance of conventional Bayesian neural networks and deep ensemble methods, particularly in terms of uncertainty estimation and out-of-distribution (OOD) detection.

**Strengths:**

* The experimental results demonstrate that the proposed method consistently improves uncertainty estimation and OOD detection.
* The proposed method is straightforward and convincing. Additionally, Figure 1 effectively provides an intuitive overview of the method.

**Weaknesses:**

* I am not yet fully convinced that the problem addressed in this paper is of breakthrough significance. While it is beneficial to develop a better way to aggregate probabilities, I believe such a method may not have a substantial impact on the Bayesian neural network (BNN) community, as deep ensembles and BNN methods already provide sufficient performance. In my view, focusing on methods that reduce inference or training time (e.g., model parameter ensemble methods) or those that scale well with large datasets (e.g., last-layer Bayesian methods) would be more important topics, as the paper itself discusses. Moreover, modern neural networks no longer suffer significantly from overconfidence [1, 2].
* Even if I acknowledge the importance of the stated problem, the performance gain is consistent but somewhat marginal. I suspect this performance gap could even decrease when larger training datasets or modern data augmentation methods are used (see Figure F1 in [3]).
* The computational cost is significantly higher than simple probability averaging, which may be a barrier to using this method in real-world applications.

In summary, while I recognize that the proposed method consistently outperforms simple probability averaging with solid evidence, in my humble opinion, the contributions are somewhat marginal. Additionally, the increased computational cost may hinder its application to real-world problems.



[1] Minderer, Matthias, et al. "Revisiting the calibration of modern neural networks." Advances in Neural Information Processing Systems 34 (2021): 15682-15694.
[2] Vishniakov, Kirill, Zhiqiang Shen, and Zhuang Liu. "Convnet vs transformer, supervised vs clip: Beyond imagenet accuracy." arXiv preprint arXiv:2311.09215 (2023).
[3] Park, Namuk, and Songkuk Kim. "How do vision transformers work?." arXiv preprint arXiv:2202.06709 (2022).

**Questions:**

Please see the weaknesses.

---

### Official Review · Reviewer_kVbE · 2024-10-31

**Soundness:** 4
**Presentation:** 4
**Contribution:** 2
**Rating:** 8
**Confidence:** 4

**Summary:**

The paper proposes a simple method to estimate epistemic uncertainty and make predictions based on sampled predictions of Bayesian Neural Networks (BNNs). For that, it determines a credal set representation using an interval defined by lower and upper bounds. These bounds are derived from sampled predictions of a BNN or Deep Ensemble (DE). Using this credal set representation, the authors compute the epistemic uncertainty and a single prediction (termed intersection probability). Their experiments demonstrate that these epistemic uncertainty estimates are more effective for out-of-distribution (OOD) detection compared to baseline BNNs and DEs. Furthermore, they show improved calibration across different distributional shifts of the intersection probability.

**Strengths:**

- The paper is really well-written and easy to follow.
- The proposed idea is really simple and is highly applicable in many settings.
- The experiments are nicely structure and follows the evaluation protocol of the literature.
- The authors clearly highlight the approach's limitations stemming from its computational complexity.

**Weaknesses:**

- **Related Work:** The approach is presented as innovative but lacks clear placement within existing literature. While there is significant discussion on credal sets in ML, the authors only briefly touch on their application in deep learning, mentioning computational complexity as a key challenge. After a short research I noticed some papers focusing on credal sets in deep learning [1, 2] and was wondering why these are not discussed in the related work. More importantly, how is this paper improving the state-of-the-art, and what are the differences to other credal set papers that are used in deep learning? Given these facts, I can not accurately determine the novelty of the proposed method.
- **Metric inconsistencies with OOD literature:** I noticed some inconsistencies in the evaluation metrics compared to related papers, particularly with AUROC values for baseline methods. For instance, Table 1 reports an AUROC of 88.63 for CIFAR-10 vs. SVHN using ResNet-18, which is significantly lower than the 93 reported in related work, including [3]. It’s unclear why this discrepancy exists—perhaps due to differences in experimental setup, data preprocessing, or other implementation details. Clarifying these points would help verify the reliability of comparisons and ensure the evaluation is aligned with the literature.
- **Hyperparameter:** The choice of the hyperparameter $J$ remains unclear; the authors simply state $J=20$ for CIFAR-100 and $J=50$ for ImageNet, without explaining the basis for these values. It would be helpful if they clarified the selection process.
Additionally, while the authors discuss the method's computational complexity and propose an approximation to mitigate it, they moved the PIA experiments to the appendix. Given the main paper includes the algorithm as a contribution, the experimental results on PIA should also appear in the main text to reflect the algorithm’s practical impact.


[1] Michele Caprio, Souradeep Dutta, Kuk Jang, Vivian Lin, Radoslav Ivanov, Oleg Sokolsky, and Insup Lee. Credal Bayesian Deep Learning. arXiv preprint arXiv:2302.09656, 2023.
[2] Wang, Kaizheng, et al. "CreINNs: Credal-Set Interval Neural Networks for Uncertainty Estimation in Classification Tasks." arXiv preprint arXiv:2401.05043, 2024.
[3] Van Amersfoort, Joost, et al. "Uncertainty estimation using a single deep deterministic neural network." ICML, 2020.

**Questions:**

Can you provide feedback regarding the stated weaknesses?

---

### Official Review · Reviewer_3c79 · 2024-11-03

**Soundness:** 4
**Presentation:** 2
**Contribution:** 3
**Rating:** 8
**Confidence:** 3

**Summary:**

This article proposes a computationally feasible method for improving the quality of Uncertainty Quantification (UQ) estimations from sample-based models (in a broader sense), such as BNNs and DEs. The proposed method utilizes credal sets to compute the entropy bounds within the probability interval. Intense experiments are conducted on a collection of datasets and models, and the proposed method shows a clear advantage over baseline methods.

**Strengths:**

- The proposed method is simple yet effective, as being demonstrated in the widely-conducted experiments.
- The proposed approximation algorithm, PIA, effectively approximates the original optimization problem and ultimately makes the idea practically tractable.
- The proposed method is easy to "wrap" around various methods as long as they provide multiple samples of model predictions.
- Experimental details are well documented.

**Weaknesses:**

Overall speaking, the paper is a bit tricky to follow as lots of details are densely presented. It will be clearer if some parts (e.g., detailed experiment setups, ablation studies on # of predictive samples, etc.) could be moved to the appendix. Some parts of the appendix, e.g., A.6 ablation studies on PIA parameter $J$, feel more important to me compared to some of the ablation studies listed in the main body.

Besides, although I am unaware of existing similar works, it would be beneficial if the authors could provide clear intuitions/reasonings on why this simple method can drastically improve baseline results, and clarify their contributions (for example, the approximation algorithm). Visualizations of real experiments can be provided to let readers grasp a general idea of what the credal sets are like in practice, as well as the prediction produced by the intersection probability transform. The paper provides a tremendous number of experiment results, which is good and validates the proposed method, yet some intuitions will still be very helpful on top of those results.

**Questions:**

As I stated above, I'd like to know the following (even simple, intuitive answers will be helpful):
- Why can the simple-looking method drastically improve the baseline results?
- How well does the proposed method perform compared to methods listed in, e.g., lines 116-125?
- Are there any visualizations/etc. of the credal sets in practice? (real-world datasets with well-trained classifier samples)

---

> ### Comment · Reviewer_3c79 · 2024-11-26
>
> Thank you for the detailed response. The visualizations and additional explanations have greatly clarified my concerns (and made the paper more convincing), yet I am still a bit unsure about the fundamental difference that makes credal wrappers outperform baseline methods. Intuitively, it is true that the credal set grows larger for OOD inputs, yet the same happens for baseline methods as samples will scatter across a larger domain within the prob simplex, leading to a larger estimation of EU. As you said, the difference between intersection probability and average prediction is clear. Yet, why does the credal set as a whole, rather than the sole intersection probability, significantly outperform baselines in estimating uncertainties? Could it be related to the advantage of “min” “max” operations in Eq. 4, compared to “mean” in Eq. 2? Or is it because of some difference between the support of ensemble predictions and the credal set? Or is it because of something else?
>
> I appreciate the comprehensive response provided by the authors, which improves the clarity of the paper. Therefore, I’d like to improve my score. Thank you for your efforts.

---

> ### Comment · Reviewer_3c79 · 2024-12-03
>
> Thank you for your additional explanations, which have addressed my concerns.

---

### Official Review · Reviewer_TRkv · 2024-11-03

**Soundness:** 3
**Presentation:** 2
**Contribution:** 3
**Rating:** 8
**Confidence:** 3

**Summary:**

This paper introduces the "intersection probability" to model averaging, to improve uncertainty estimation in OOD detection. This introduced method is an alternative for aggregating predictive distributions by different models/particles, and considers the lower/upper bounds of the probability interval.

**Strengths:**

1. The idea of using probability intervals to reflect the epistemic uncertainty is appealing.

2. The intersection probability transformation has a deterministic solution, and thus it is as efficient as other alternatives in model averaging.

3. Experimental results are strong and comprehensive.

**Weaknesses:**

1. Except for table results, I expect a more direct demonstration of why intersection probability is better than naive averaging. For example, can you visualize the uncertainty on simple regression tasks?

2. Computing epistemic uncertainty using probability intervals is costly. I expect discussions on the extra computational cost compared with entropy.

**Questions:**

Please refer to weaknesses.

---

> ### Comment · Reviewer_TRkv · 2024-11-25
>
> Thank you for the comprehensive responses and sorry for the late reply.
>
> I am totally convinced by Fig. R1 and R2 that the intersection probability is better, and I recommend moving R1/R2 to the main body if you still have enough space.
>
> The discussion of computational cost is fair. My concern has been addressed.
>
> I am happy to raise my score.

---

### Meta-Review · Area_Chair_VcDg · 2024-12-23

**Metareview:**

The paper introduces a new approach called the "credal wrapper" to improve uncertainty estimation in classification tasks using Bayesian neural networks and deep ensembles. The method constructs a probability interval per class and maps that interval to a credal set.  The authors show in experiments that their method maintains high accuracy while being better calibrated than the baselines, and that the method is effective for performing out-of-distribution detection.

The reviewers mostly agreed to accept the paper although one reviewer thought it was borderline reject (8, 8, 8, 5).  The dissenting reviewer seemed to find the approach sensible, but mostly voiced concerns over the impact of the paper / method.  They seemed to suggest that existing methods were sufficient to improve the calibration of deep neural networks, or that modern networks are well-calibrated already, and that the current goal of the literature should be to improve efficiency.  The reviewers agreed the paper was well-written, the approach novel, simple and sensible and the experiments convincing.  As weaknesses, the reviewers noted lack of comparison to existing literature, limited theory, and high computational cost.

Overall, the reviewers seem to find this a strong paper.  There doesn't seem to be a reason to overrule this assessment.  While the criticism of the dissenting reviewer is noted (there is an extensive literature on improving calibration already), I would disagree that this problem seems at all solved by existing tools such as ensembling (or that existing models are well-calibrated).  Therefore the recommendation is to accept.

**Additional Comments On Reviewer Discussion:**

The authors responded extensively to all reviews during the rebuttal period.  For the most part, the rebuttals were quite well received and two reviewers raised their scores.  The reviewer who gave a rating of 5 acknowledged the authors' response but decided to maintain their score.  In my opinion, this reviewers contention is subjective as it pertains to the perceived impact of the work.  This is not really an issue, but it therefore would be difficult for the authors to address in a rebuttal.

---

### Decision · Program_Chairs · 2025-01-22

Accept (Spotlight)